# Modeling memory in gravel-bed rivers: A flow history-dependent relation for evolving thresholds of motion

Claire C. Masteller[1], Joel P.L. Johnson[2], Dieter Rickenmann[3], and Jens M. Turowski[4]

[1]Department of Earth, Environmental, and Planetary Sciences, Washington University in St. Louis.
[2]Jackson School of Geosciences, University of Texas at Austin.
[3]WSL Swiss Federal Institute for Forest, Snow, and Landscape Research.
[4]Helmholtz Centre Potsdam, GFZ German Research Centre for Geosciences

*Correspondence to*: Claire C. Masteller (cmasteller@wustl.edu)

**Abstract.** Thresholds of motion ($\tau_c^*$) strongly control bedload transport in gravel-bed rivers. Uncertainty in $\tau_c^*$ limits the accuracy of predictions of transport and morphologic change. To improve our quantitative understanding of morphodynamic feedbacks in rivers, we propose a flow history-dependent model where $\tau_c^*$ evolves temporally as a function of bed shear stress. Relatively low shear stresses strengthen the bed, increasing $\tau_c^*$ and reducing transport. Larger floods rapidly weaken the bed, decreasing $\tau_c^*$ and increasing transport. We calibrate the model to a 23-year record of flow and bedload transport from the Erlenbach Torrent, Switzerland, and find that the model predicts the field-based $\tau_c^*$ record more accurately than assuming a constant $\tau_c^*$. Calibrated parameters describing strengthening are more tightly distributed than weakening parameters, which suggests that magnitudes of bed weakening may be more variable and difficult to accurately predict as a function of flood characteristics than bed strengthening during lower flows.

## 1 Introduction

Erosion, deposition and morphological change in gravel-bed rivers result from bedload transport. Gravel transport rates are notoriously difficult to accurately predict in natural rivers because they are influenced by a wide variety of factors including water discharge, local channel morphology, and upstream sediment supply. These factors can vary both spatially and temporally. Nonetheless, most equations to predict bedload flux simplify this complexity using a deterministic parameterization which only considers flow intensity and a transport threshold. For example, the classic Meyer-Peter and Müller (1948) equation (MPM) can be expressed simply as $q_s^* = 4(\tau^* - \tau_c^*)^{1.5}$ for $\tau^* \geq \tau_c^*$ (Wong and Parker, 2006), where $q_s^*$ is dimensionless sediment flux per unit channel width (Einstein number), $\tau^*$ is nondimensional shear stress (Shields stress), and threshold parameter $\tau_c^*$ is the critical Shields stress. Shields stress is $\tau^* = \tau/(\rho_s - \rho)gD$, where $\tau$, $\rho_s$, $\rho$, $g$ and $D$ are dimensional bed shear stress (Pa), sediment density (kg/m³), water density (kg/m³), gravitational acceleration (m/s²), and median sediment diameter (m). Einstein number is defined as $q_s^* = q_s/\sqrt{(\rho_s/\rho - 1)gD^3}$, where $q_s$ is the volumetric bedload

transport rate per unit width (m³/s/m). While the threshold parameter ($\tau_c^*$) describes the nondimensional shear stress at the onset of sediment motion, it is also often used as a physically meaningful, but empirically determined, fitting parameter in bedload transport models (e.g., Engelund & Fredsoe, 1976; Luque & Beek, 1976; Meyer-Peter & Müller, 1948; Wong & Parker, 2006; Shields, 1936; Wiberg & Smith, 1987). For bedload equations like the MPM, $\tau_c^*$ is a lumped parameter that implicitly accounts for all of the factors that influence bedload transport rates apart from $\tau^*$.

35       Even during large floods, bedload transport often occurs at shear stresses only slightly exceeding threshold conditions, making transport rate predictions particularly sensitive to threshold values (e.g. Parker, 1978; Phillips & Jerolmack, 2016; Phillips et al., 2022; Pretzlav et al., 2020). Transport thresholds also strongly influence gravel-bed channel reach morphologies (e.g. Parker, 1978; Phillips et al., 2022) and modulate the mapping of climate onto fluvial processes, informing short- and long-term sediment fluxes and the relative importance (or unimportance) of extreme events for channel evolution (e.g. Blom
et al., 2017; DiBiase & Whipple, 2011; Lague et al., 2005; Shobe et al., 2018; Tucker & Bras, 2000). For these reasons, improving our ability to calculate thresholds of motion is critical not only for predicting transport rates but also for predicting mountain river morphodynamics and channel stability.

      Early work assumed that thresholds were primarily controlled by grain weight relative to the fluid. By accounting for these variables through nondimensionalization, $\tau_c^*$ was initially thought to be approximately constant for typical conditions in
gravel-bed rivers (e.g. Buffington & Montgomery, 1997; Shields, 1936). However, Buffington & Montgomery (1997) showed that $\tau_c^*$ varied systematically with the ratio of the median grain size to flow depth, independent of shear stress. More recent work has explored how both flow and grain interactions lead to inherent variability in $\tau_c^*$. Thresholds vary spatially with the surrounding grain size distribution (Parker, 1990), reach slope (Lamb et al., 2008; Mueller et al., 2005), bed morphology (Monsalve & Yager, 2017; Powell & Ashworth, 1995; Roberts et al., 2020), and changes in riverbed microtopography
(Brayshaw, 1985; Hodge et al., 2019; Kirchner et al., 1990; Masteller & Finnegan, 2017; Yager et al., 2018).

      Thresholds for motion also evolve over time. For example, hysteresis in bedload transport rates is often observed between the rising and falling limbs of individual floods (Hsu et al., 2011; Mao, 2018; Mao et al., 2014; Pretzlav et al., 2020; Reid et al., 1985; Roth et al., 2017). Predicting hysteresis using the MPM and similar bedload models requires the threshold parameter to evolve over the course of a flood event (assuming that the prefactor and exponent remain constant). Changes in

$\tau_c^*$ over multiple events have also been observed, and in most cases $\tau_c^*$ values remain correlated across events, indicating a memory of past conditions (Downs & Soar, 2021; Hassan et al., 2020; Johnson, 2016; Lenzi et al., 2004; Mao, 2018; Masteller et al., 2019; Rickenmann, 2018, 2020; Saletti et al., 2015; Turowski et al., 2011).

     Variable flow strength influences threshold evolution through time. Reid et al. (1985) first suggested the influence of antecedent flows based on field-based bedload transport monitoring, hypothesizing that longer inter-flood durations led to

increases in $\tau_c^*$ and reduced sediment transport rates. Experiments have confirmed that the magnitude of inter-event flow affects $\tau_c^*$ evolution (Haynes & Pender, 2005; Masteller & Finnegan, 2017; Monteith & Pender, 2005; Ockelford et al., 2019; Ockelford & Haynes, 2013; Paphitis & Collins, 2005). With little to no active sediment transport, grain-scale changes in interlocking and surface reorganization increase particle resistance to motion (Masteller & Finnegan, 2017; Ockelford & Haynes, 2013; Yager et al., 2018). Pretzlav et al. (2020) documented systematic discharge-dependent increases and decreases

in motion thresholds and associated diurnal transport hysteresis during several weeks of snowmelt flooding, using instrumented "smartrocks" to measure transport. Reduction of $\tau_c^*$ following larger floods has been attributed to significant reorganization of the riverbed (Lenzi et al., 2004; Turowski et al., 2009).

     Sediment supply also influences threshold evolution. Hysteresis can be caused by sediment supply variations through time (Moog and Whiting, 1998; Mao et al., 2014), changing thresholds. Increased sediment supply from channel banks and

hillslopes can be important in destabilizing the bed surface or introducing mobile unconsolidated material, reducing thresholds of motion (Turowski et al., 2011; Recking et al., 2012; Rickenmann, 2020). Building on observations by Recking et al. (2012), Johnson (2016) developed a model in which $\tau_c^*$ evolves as a function of net erosion or deposition, which are controlled by sediment supply in relation to transport capacity. After calibration to laboratory experiments, the evolving-$\tau_c^*$ model successfully predicted how transport rates responded to pulses in sediment supply.

Decades of monitoring data from the Erlenbach torrent in Switzerland similarly provide evidence for transport thresholds evolving with both sediment supply and discharge variability. Rickenmann (2020) showed that variations in sediment availability on the bed correlated with sediment transport rate fluctuations and evolving thresholds, suggesting that thresholds depend on upstream sediment supply. For the same stream, Masteller et al. (2019) showed that the magnitude of antecedent flows also influenced the evolution of $\tau_c^*$ for individual years. Consistent with experiments, Masteller et al. (2019)

observed that the start of transport events showed increases in critical Shields stress with increasing inter-event flow magnitude (herein termed "strengthening") for an intermediate range of flows spanning inter-event periods and floods with observable sediment transport. However, following even higher-magnitude flows, the threshold for motion decreased (herein termed "weakening"). Masteller et al. (2019) hypothesized that the transition from bed strengthening to bed weakening was associated with a transition from local rearrangement of particles to more intense transport disrupting bed structure via particle collisions,
and/or enhanced upstream sediment supply through upstream bed erosion (Yager et al., 2012), and/or enhanced hillslope-channel coupling (Golly et al., 2017). Thus, both flow strength and sediment supply likely influence thresholds of motion in the Erlenbach torrent (Rickenmann, 2020; Masteller et al., 2019; Turowski et al., 2011).

The ability to accurately predict threshold evolution through time—thereby improving bedload transport rate predictions—remains elusive due to a lack of validated models. Equations have been proposed to describe temporal bed
strengthening as a function of the duration of bed exposure to a constant, inter-event flow magnitude and an initial $\tau_c^*$ based on experimental data (e.g. Ockelford et al., 2019; Paphitis and Collins, 2005). However, because these models only focus on inter-event strengthening effects, they cannot capture decreases in $\tau_c^*$. Johnson's (2016) model predicts $\tau_c^*$ evolution as a function of changing sediment supply. Nonetheless, this model is an incomplete description of $\tau_c^*$ evolution because it does not account for riverbed strengthening or weakening directly caused by the flow. Notably, to our knowledge, none of these
equations have been used to describe field observations of temporally varying $\tau_c^*$.

Our goals in the present work are (i) to propose a new model in which $\tau_c^*$ evolves as a function of flow magnitude and encapsulates some memory of past shear stresses as reflected in the changing state of the riverbed, and (ii) to evaluate whether this discharge-dependent model can capture annual strengthening and weakening trends observed in Erlenbach field data (Masteller et al., 2019). While sediment supply variations can also influence threshold evolution in general (Johnson,
2016), and in the Erlenbach in particular (Rickenmann, 2020), we explore how well threshold evolution can be predicted using only a timeseries of river discharge.

## 2 Model Development

Johnson (2016) argued that $\tau_c^*$ is a "state variable" for gravel-bed river morphodynamics because it simultaneously controls transport rates and evolves due to feedbacks with fluid shear stresses and transport rates. Our new equations take a
similar form to Johnson (2016), where changes in $\tau_c^*$ depend not only on discharge-dependent shear stress, but also on the current state of the transport system as characterized by $\tau_c^*$ itself. The rate of change of $\tau_c^*$ depends on two terms, which both evolve as a function of the transport capacity, $\tau^*/\tau_c^*$. Conceptually, the first right hand-side term (starting with $k_1$) represents strengthening processes that increase $\tau_c^*$, while the second term (starting with $k_2$) represents weakening processes that reduce $\tau_c^*$:

$$\frac{\partial \tau_c^*}{\partial t} = k_1 B \left(1 + \left(\frac{\tau^*}{\tau_c^*}\right)^{-\gamma}\right)^{-1} - k_2 (1-B)\left(\frac{\tau^*}{\tau_c^*} - 1\right)^{\varepsilon} H[\tau^*/\tau_c^* - 1], \tag{1}$$

where $t$ is time, and

$$B = \frac{\tau_{cmax}^* - \tau_c^*}{\tau_{cmax}^* - \tau_{cmin}^*} \quad \text{for } \tau_{cmin}^* < \tau_c^* < \tau_{cmax}^*. \tag{2}$$

Scaling factor $k_1$ and exponent $\gamma$ influence the form and magnitude of the strengthening term, while $k_2$ and $\varepsilon$ do the same for the weakening term (Fig. 1); these parameters are empirically calibrated below. Both $k_1$ and $k_2$ have units of 1/time, where the
units of time will depend on the timestep of the discharge timeseries used as model input. $H$ is the Heaviside step function ($H[\tau^*/\tau^*_c-1]=0$ for $\tau^*/\tau^*_c<1$; $H[\tau^*/\tau^*_c-1]=1$ for $\tau^*/\tau^*_c>=1$) such that weakening only occurs when transport occurs ($\tau^* > \tau_c^*$) (Fig. 1A). $B$ is called the "feedback parameter," described below. $\tau_{cmin}^*$ and $\tau_{cmax}^*$ are upper and lower bounds imposed on $\tau_c^*$, representing physical limits for how loosely-packed and mobile, or tightly-packed and immobile, the bed surface can become (Johnson, 2016).

The strengthening and weakening terms combine to cause increases and decreases in $\partial \tau_c^*/\partial t$ (Eq. 1; Fig. 1). The strengthening term is generally sigmoidal for $\gamma > 1$; it goes to zero as $\tau^*$ approaches zero, and asymptotes to a value of $k_1 B$ for $\tau^*/\tau_c^* \gg 1$ (Fig. 1a). We chose a sigmoidal form to allow for strengthening over a wide range of flows, but also limit the amount of incremental strengthening that can result from changes in grain organization at higher transport capacities. When

$\tau^*/\tau_c^* < 1$, flow causes the bed to become stronger but not weaker, consistent with previous observations (Haynes & Pender,

2005; Masteller et al., 2019; Masteller & Finnegan, 2017; Monteith & Pender, 2005; Ockelford et al., 2019). Strengthening

increases as $\tau^*/\tau_c^*$ approaches 1, consistent with some (Paphitis & Collins, 2005) but not all previous work (Haynes & Pender,

2007). Strengthening increases further for $\tau^*/\tau_c^* > 1$, consistent with protrusion-dependent thresholds (Masteller and

Finnegan, 2017, Yager et al., 2018; Masteller et al., 2019), and with coarse grain clustering which increases bed stability and

requires transport to develop (Brayshaw, 1985; Church et al., 1998; Hassan et al., 2020; Johnson, 2017; Strom et al., 2004).

130            At the same time, as $\tau^*/\tau_c^*$ exceeds 1, the weakening term becomes increasingly important (Eq. 1; Fig. 1). In the

absence of other constraints on the functional form of weakening with increasing $\tau^*/\tau_c^*$, we chose a power-law relation for

simplicity. It seems likely to us that beds rapidly lose their strength as transport rate increases and fewer grains are interlocked

through intergranular friction (Yager et al., 2018). Higher shear stresses capable of mobilizing more sediment grains can

destabilize a larger fraction of the bed. Impacts from transported grains may also directly contribute to destabilization (Ancey

& Heyman, 2014; Heyman et al., 2014; Lee & Jerolmack, 2018; Martin et al., 2014). Nonetheless, we note that Equation 1 is

agnostic towards any specific processes driving strengthening and weakening. The combination of terms results in the

transition from strengthening to weakening occurring at different $\tau^*/\tau_c^*$, depending on γ, ε, $k_1$, $k_2$, and $\tau_c^*$ (Fig. 1b).

        $B$ is considered a "feedback parameter" because it contributes to $\partial\tau_c^*/\partial t$ being a function of $\tau_c^*$ (Johnson, 2016). $B$

has a value between 0 and 1, and changes the importance of the strengthening and weakening terms, depending on the current

value of $\tau_c^*$ relative to $\tau_{cmin}^*$ and $\tau_{cmax}^*$. A loosely-packed bed, with $\tau_c^*$ close to $\tau_{cmin}^*$ and $B$ close to 1, can strengthen

significantly in a low flow (increasing $\tau_c^*$), but a high flow would not cause a significant decrease in $\tau_c^*$ because the bed is

already relatively weak. Conversely, a bed that was nearly as tightly-packed as physically possible, with $\tau_c^*$ close to $\tau_{cmax}^*$ and

$B$ close to 0, will minimally increase $\tau_c^*$ in response to a low flow, but a destabilizing flood would cause a significant decrease

in $\tau_c^*$ (Johnson, 2016).

145            To estimate the range of possible $\tau_{cmin}^*$ and $\tau_{cmax}^*$ values, we use a compilation of $\tau_c^*$ as a function of channel reach

slope (Johnson, 2016; Lamb et al., 2008; Prancevic & Lamb, 2015). A best-fit power-law regression of the compiled data gives

$\tau_c^* = 0.42S^{0.7} + 0.03$, with $R^2 = 0.41$ where $S$ is channel reach slope (an approximation of energy or water surface slope).

The constant is included so that the function asymptotes to a physically reasonable value (i.e., $\tau_c^* = 0.03$) as slope approaches zero. Field and flume data were weighted equally in the nonlinear best-fit regression, removing possible bias from there being ~3.5 times more flume data points. We then visually determined the following minimum and maximum bounds to accommodate almost all compilation data points, assuming the same best-fit exponent (0.7):

$$\tau_{cmin}^* = 0.14S^{0.7} + 0.0075 \tag{3}$$

$$\tau_{cmax}^* = 1.4S^{0.7} + 0.075 \tag{4}$$

These empirical $\tau_{cmin}^*$ and $\tau_{cmax}^*$ relations capture the slope-dependence of gravel thresholds of motion compiled in both flume and field settings. Equations 1 and 2 assume that $\tau_{cmin}^*$ and $\tau_{cmax}^*$ are limits that the threshold can approach but does not reach due to the feedbacks implemented by the B parameter. Equations 1 and 2 are only defined between these bounds. We acknowledge that many factors beyond slope may influence the possible range of $\tau_c^*$ at a given site, including grain size distributions (e.g., Parker, 1990) and differences in relative roughness (Buffington and Montgomery, 1997; Schneider et al., 2015). In the absence of independent constraints on $\tau_{cmin}^*$ and $\tau_{cmax}^*$ that could be used to describe a particular field site, the compilation should reasonably represent the range of possible values.

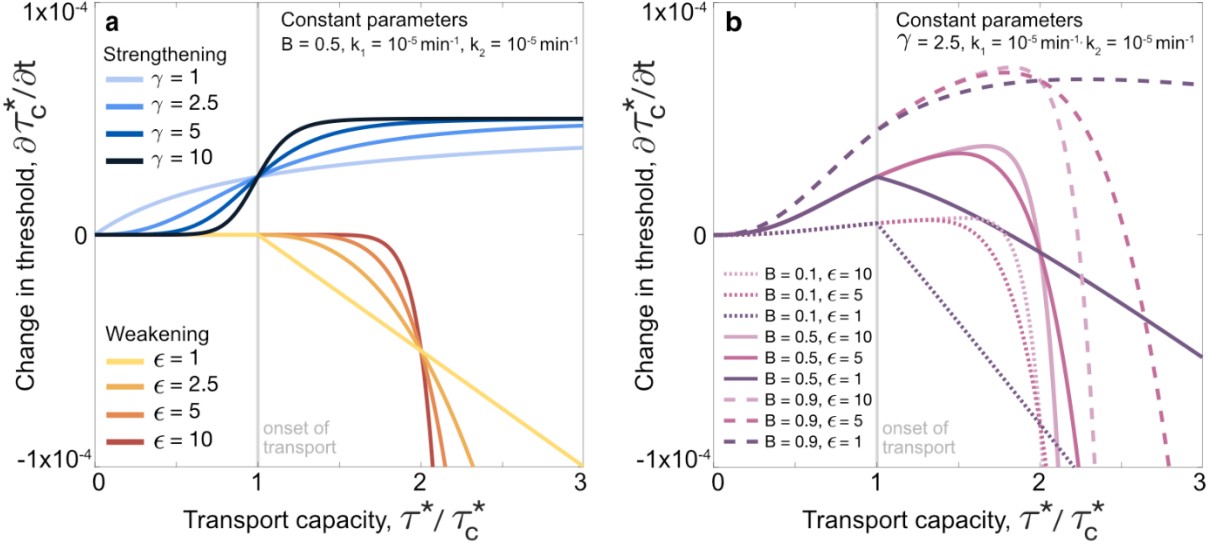

**Figure 1: a)** Predicted change in threshold for strengthening (blue) and weakening terms (red) for a range of $\gamma$ and $\varepsilon$ values. $k_1$, $k_2$, and $B$ are constant. **b)** Predicted change in threshold for the full model for different $B$ and $\varepsilon$. $k_1$, $k_2$, and $\gamma$ are held constant. This example uses Erlenbach values calculated as $\tau^*_{cmin} = 0.036$, $\tau^*_{cmax} = 0.36$ using Equations 3 and 4.

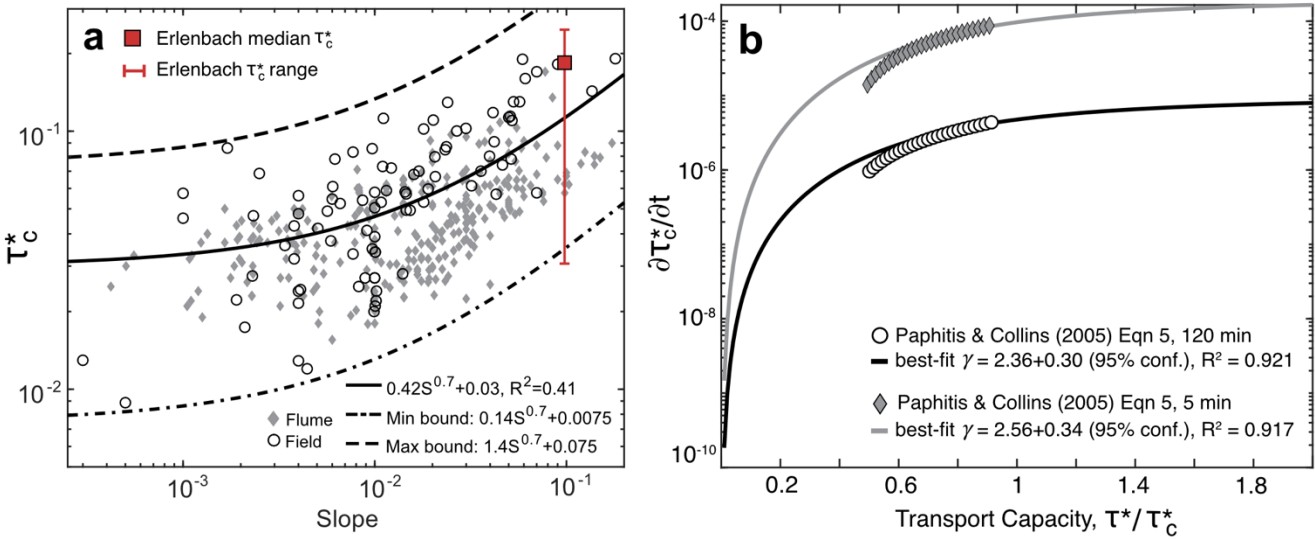

**Figure 2. a)** Compiled $\tau^*_c$ data as a function of slope from both field studies (grey diamonds) and flume experiments (open circles). Range of $\tau^*_c$ from the Erlenbach field site is plotted in red. Compiled data are limited to slopes $S < 0.2$ and median grain sizes $D_{50} \geq$ 2 mm. Most data were compiled by Prancevic and Lamb (2015), building on Buffington and Montgomery (1997), with additional data from Olinde (2015) and Lenzi et al. (2006). **b)** Calibration of strengthening term exponent, $\gamma$, based on Paphitis and Collins (2005) for both the shortest conditioning time (5 minutes) and longest conditioning time (120 minutes) spanned by the Paphitis and Collins (2005) data. Regressions to their best-fit empirical equation give gamma exponents within uncertainty of each other.

We calibrated $\gamma$ using experimental data from Paphitis and Collins (2005) to reduce the number of free parameters in the model (i.e. $\gamma$, $\varepsilon$, $k_1$, $k_2$). Paphitis and Collins (2005) conducted experiments using fine, medium and coarse sand, in which they systematically varied both the conditioning time ($E_D$, the duration of flow below the threshold condition) between 5 and 120 minutes, and the ratio of shear velocity ($u_\tau$) to initial critical shear velocity ($u_{\tau ci}$). The ratios of shear velocities they explored was between 70% and 95% of critical, which corresponds to conditioning flow shear stresses between 50% and 90% of the initial critical shear stress. Based on their experimental results, they presented the following equation to describe their data:

$$\frac{u_{\tau c(t)}}{u_{\tau ci}} = 1.05\left[1 - 0.01e^{(-0.005E_D)}\right] + \left[0.005 + 0.1\left(\frac{u_\tau}{u_{\tau ci}} - 0.7\right)\right]ln(E_D) + 0.06\left[10^{-7\left(0.97 - \frac{u_\tau}{u_{\tau ci}}\right)}\right] \quad (5)$$

for $0.7 \leq \frac{u_\tau}{u_{\tau ci}} \leq 0.95$ and $E_D \leq 120$ minutes, where $u_{\tau c(t)}$ is the critical shear velocity following low-flow conditioning.

Paphitis and Collins (2005) reported that this function fit their experimental data with a correlation coefficient of 0.83 (i.e. $R^2$=0.69).

     To calibrate $\gamma$, we calculate $u_{\tau c(t)}/u_{\tau ci}$, for a range of $E_D$ and $u_\tau$, using values of $u_{\tau ci}$ reported for their coarse sand

(D=0.774 mm) experiments, corresponding to an initial critical shear velocity of $u_{\tau ci}$=0.0195 m/s. We then square $u_{\tau c(t)}/u_{\tau ci}$

to convert to a Shields stress ratio (i.e., $\left(u_{\tau c(t)}/u_{\tau ci}\right)^2 = \tau_c/\tau_{ci} = \tau_c^*/\tau_{ci}^*$, where the $i$ subscript indicates the initial value, and

$\tau_c$ and $\tau_c^*$ evolve through time). We then numerically calculate the partial derivative of Equation 5 with respect to time, $\partial \tau_c^*/\partial t$.

Figure 2B shows a nonlinear regression (using Matlab's cftool) of the strengthening term in our model (Equation 1) to $\partial \tau_c^*/\partial t$

calculated from Equation 5. This regression provides a best-fit estimate of $\gamma$, including empirical regression uncertainties. For

below-threshold conditions explored in the Paphitis and Collins (2005) experiments, the weakening term in Equation 1 is zero.

Given this, the calibration of $\gamma$ described here is not influenced by other model parameters, in particular, the weakening

exponent $\varepsilon$. Our reported 95% confidence interval on $\gamma$ only represents the empirical regression uncertainty when fitting our

function to Equation 5. Therefore, it is likely that a somewhat wider range of $\gamma$ may be able to fit the range of the experimental

data from Paphitis and Collins (2005), and the true range of possible values may be somewhat larger than 2.5+/-0.32.

     We use the data and fitting function of Paphitis and Collins (2005) to calibrate $\gamma$ because it is the most complete and

internally consistent dataset that we are aware of with sufficient constraints to describe the evolution of $\tau_c^*$ as a function of

both transport capacity and time. Nonetheless, a possible limitation of applying these experimental data to calibrate our model

is that the Paphitis and Collins (2005) experiments were conducted with unimodal sand. Specifically, boundary Reynolds

numbers in their experiments are transitional between hydraulically smooth and hydraulically rough flow. For the coarsest

grains they use ($D_{50}$ = 0.0774 mm), boundary Reynolds number $Re_w = u_\tau k_s/v \approx 15$ , where $k_s$ is a roughness length scale

assumed to be $D_{50}$, and $v$ is the kinematic viscosity of water. If $k_s$ was instead assumed to be a multiple of $D_{50}$ (such as $k_s$ =

3.5$D_{84}$) then $Re_w$ would be closer to the hydraulically rough flow criteria of $Re_w \geq 100$. It is also worth noting that grain size

did not explicitly factor into Equation 5, beyond its implicit control on $u_{\tau ci}$. The insensitivity of their results to grain size

suggests that the results may not depend significantly on grain size or on hydraulically rough flow being fully developed.

Further, converting critical shear velocities from their coarse sand experiments to critical Shields stress yields $\tau_c^* \approx 0.031$,

consistent with typical critical Shields stresses reported at low slopes (consistent with their experiments) for gravel river data sets (Fig. 2a).

## 3 Field application

The Erlenbach is a small (0.7 km$^2$ watershed), steep (10% grade) channel in the Swiss Prealps. Bedload transport has been actively monitored for over 30 years by a variety of methods (Beer et al., 2015; Rickenmann, 2020; Rickenmann et al., 2012; Rickenmann and McArdell, 2007). Previous analyses have shown that the threshold for motion varies over an order of magnitude across the span of the record (Masteller et al., 2019; Turowski et al., 2011). Masteller et al. (2019) demonstrated that seasonal trends in $\tau_c^*$ were unlikely to be random, and that threshold evolution depends, in part, on the magnitude of past flows.

Our goal is to evaluate how well discharge-dependent Shields stress variations alone (Equations 1 and 2) capture first-order seasonal trends in evolving $\tau_c^*$ from well-constrained field data. We utilize publicly available 23-year, 10-minute interval discharge and bedload transport records from the Erlenbach (Rickenmann et al., 2020). To calculate thresholds of motion for model comparison, we measure the discharge when the in-channel impact plate system registers grain collisions near the beginning of a flow event, following Turowski et al. (2011). Thus, the threshold data are independent of any bedload transport model. Using a rating curve from discharge to shear stress developed by Yager (2006) and a median grain size, D$_{50}$ = 8 cm (Wyss et al. 2016), the flow and transport time series were nondimensionalized to Shields stress. The median grain size is assumed to have not changed systematically across the Erlenbach record (as discussed by Masteller et al., 2019), although hillslope sediment supply may cause grain size variability both during and in between transporting events.

Critical Shields stresses at the start of transport vary by almost an order of magnitude in $\tau_c^*$ (0.03 to 0.26). The temporal variability in $\tau_c^*$ observed at the Erlenbach is equivalent to the full range of $\tau_c^*$ observed in flume and field data for equivalent slopes (Fig. 2A). Strengthening (i.e., a systematic increase in $\tau_c^*$ for at least some portion of a given year) is dominant in 10 of the 23 years (see Masteller et al., 2019), weakening is dominant in 3 years (1992, 2014, 2015), while the remaining 10 years show both behaviors (Fig. 3e-h).

## 4 Model parameterization and application

We implement the model separately to each year's flow time series, from the first transporting event in the spring through the fall (following Masteller et al., 2019). We do not calibrate $\tau_c^*$ to the single continuous multi-year discharge and transport record because the bulk of landsliding occur during the winter months, supplying largely unconstrained amounts of sediment to the channel bed from hillslope processes (Schuerch et al., 2006). Hillslope sediment supply variations also occur during the rest of the year and likely influence thresholds and transport rates both during and in between the transporting events we consider (e.g., Rickenmann, 2020; 2024). As possible evidence of sediment supply effects during inter-event periods, Turowski et al. (2011) found that threshold discharges were often, though not always, lower at the start of a given flow event compared to the discharge at the cessation of bedload transport at the end of the previous event. This inter-event weakening cannot be captured by the model. Because the model cannot predict every trend in the field data, we focus on the start of events only to evaluate how well discharge variations alone can improve transport predictions over seasons, consistent with the analysis of Masteller et al. (2019). Future analyses could focus on threshold evaluation and model calibration during individual flood events.

Equation 1 has four free parameters: $k_1$, $k_2$, $\gamma$, and $\varepsilon$. We assign $\gamma = 2.5$ based on our calibration to Paphitis and Collins (2005) (Fig. 2b). This leaves three parameters that require calibration. For each year, we explored a range of parameter combinations for $k_1$ ($1 \times 10^{-2}$ to $1 \times 10^{-6}$ min$^{-1}$, n = 40), $k_2$ ($1 \times 10^{-2}$ to $1 \times 10^{-6}$ min$^{-1}$, n = 40), and $\varepsilon$ (1 to 10, n = 10). $k_1$ and $k_2$ were varied with log-spacing to explore all orders of magnitude equivalently. For each year of the dataset, we ran 16,000 forward simulations, reflecting all unique parameter combinations of $k_1$, $k_2$, and $\varepsilon$. For each year, we assign an initial $\tau_c^*$ value as equal to the observed $\tau_c^*$ at the first transport event and calculate changes in $\tau_c^*$ based on 10-minute discharge data, until the end of the final observed transport event.

We determined the best-fit parameter combinations that minimized Mean Absolute Error (MAE) to $\tau_c^*$ data for each year. We use MAE (rather than RMSE) to reduce the influence of any single large difference between the continuous model predictions and field-based data points which exhibit large amounts of scatter. The field data points only represent the discrete start of each event. "Annual" calibrations represent the best-fit parameters for each year. The "combined" calibration represents the single best-fit $k_1$, $k_2$, and $\varepsilon$ values which minimize MAE when MAE is averaged across all sample years, with each year

weighted equally. We compare both the annual and combined best-fit model to a constant $\tau_c^*$ (mean Erlenbach $\tau_c^* = 0.1548$, SE = 0.0014).

## 5 Results


The annual calibrations show that, for all 23 years, Equation 1 provides a better fit to the data (lower MAE) than the mean Erlenbach threshold, $\tau_c^* = 0.1547$ (Fig. 3a). Annual best-fit MAE ranges from 0.0046 (2015) to 0.0293 (1990). The mean MAE = 0.0149 from the annual calibrations is less than the mean MAE = 0.0254 when applying a constant $\tau_c^* = 0.1547$. Annual best-fit values for $k_1$ ranged from $1 \times 10^{-6}$ to $2.73 \times 10^{-5}$ min$^{-1}$ with a mean value of $6.55 \times 10^{-6}$ min$^{-1}$ (Fig. 3b). In contrast, annual

best-fit $k_2$ values spanned the entire parameter range ($1 \times 10^{-6}$ to $10^{-2}$ min$^{-1}$) with mean $k_2 = 2.63 \times 10^{-4}$ min$^{-1}$ (Fig. 3c). Best-fit annual $\varepsilon$ values also spanned the full range of parameter values explored (1 to 10) (Fig. 2c), with mean $\varepsilon = 6$ when calculated with each year weighted equally.

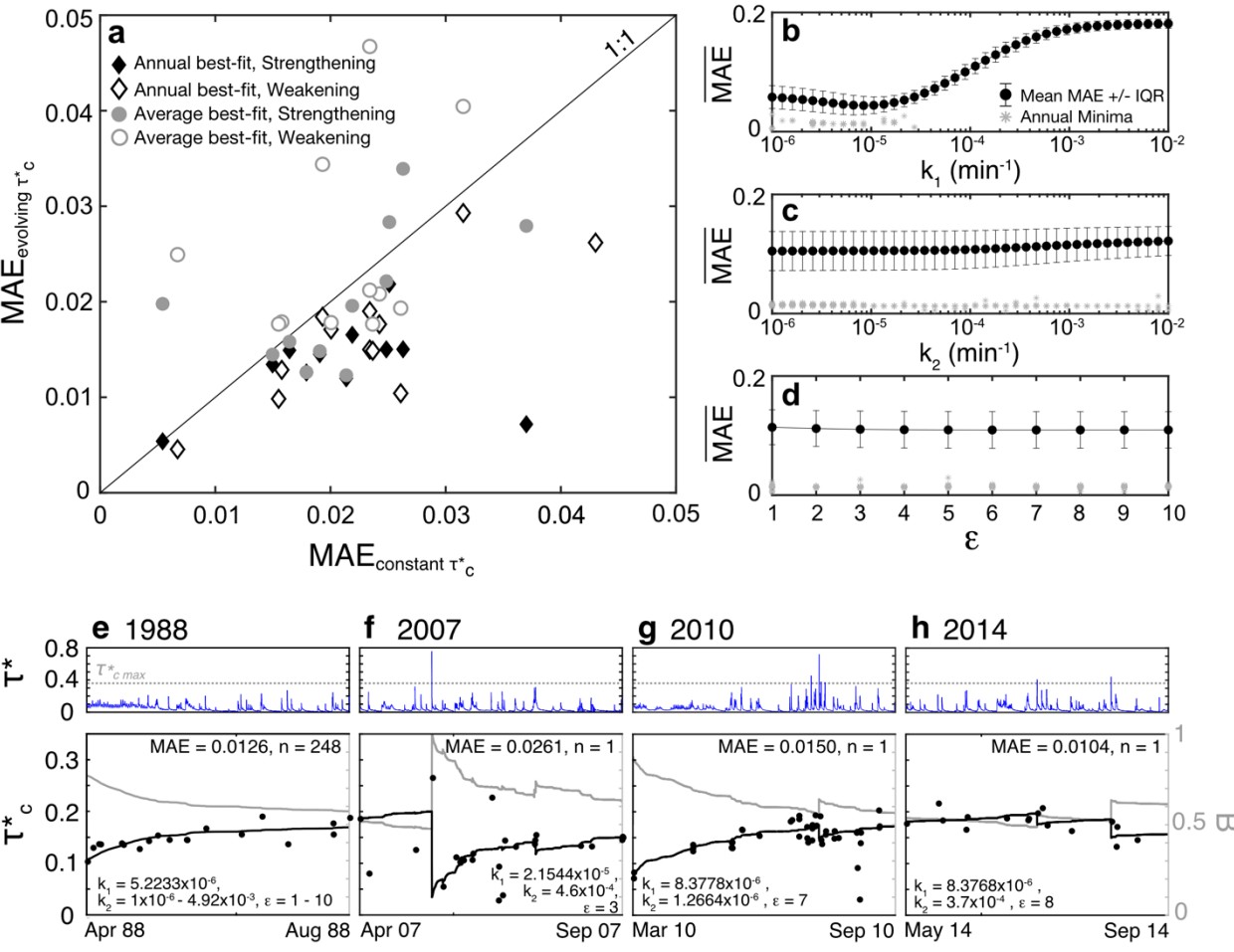

Figure 3: a) Comparison of MAE fits for constant and evolving $\tau_c^*$ values compared to $\tau_c^*$ data from the Erlenbach for both annually calibrated and combined calibration models. (b-d) Median and interquartile range of MAE for model runs binned by (b) $k_1$, (c) $k_2$, and (d) $\varepsilon$ parameters. All annual minimum MAE values and associated parameters indicated by grey stars. In the event of multiple parameter combinations yielding the minimum MAE (see discussion), all parameter combinations are plotted. (e-h) Comparison of field data (black dots represent $\tau_c^*$ at the start of each transporting event) to the best-fit model(s). Example years with (e) strengthening only (1988), (f) dominantly weakening, with strengthening as well (2007), (g) dominantly strengthening, with weakening as well (2010), and (h) dominantly weakening (2014). Upper panels show the flow time series for each year in blue; model parameter $\tau_{cmax}^* = 0.36$ is also indicated. Feedback parameter B is shown for each year (right-hand y axis). "n" gives the number of models (i.e., parameter combinations) that minimize MAE and provide an equivalent best fit to the data. Annual best-fit parameters are also specified.

The single best-fit value from the combined MAE analysis was also $\varepsilon = 6$, with best-fit $k_1 = 5.22 \times 10^{-6}$ min$^{-1}$ and $k_2 = 4.12 \times 10^{-6}$ min$^{-1}$. These best-fit $\varepsilon$, $k_1$, and $k_2$ values are consistent when averaging MAE across all years and when only averaging for the eight years with predicted weakening behavior (defined as years with step drops in $\tau_c^*$ > 2%). Model runs using the combined best-fit parameters only perform better than the constant $\tau_c^*$ assumption in 12 out of 23 sample years (mean MAE = 0.217) (Fig. 3a). Both annual and combined model performance is generally better for years which only have $\tau_c^*$ strengthening,

as evidenced by annual mean MAE = 0.0135 (combined mean MAE = 0.0202) for these years compared to years with observed weakening (annual mean MAE = 0.0163; combined mean MAE = 0.0301). In 11 years, multiple combinations of parameters resulted in the same minimum MAE value. Most of these years only have systematic strengthening, not weakening, so models are relatively insensitive to $k_2$ and $\varepsilon$ values, allowing a range of best-fit model parameterizations. The field data tend to be more variable in years with weakening; nonetheless the calibrated model captures first-order, annual trends across a range of

scenarios (Fig. 3e-h, Fig. S1). Across all examples, the dominance of the strengthening effects is demonstrated by B>0.5.

Comparison of mean MAE values of all model runs as a function $k_1$, $k_2$, and $\varepsilon$ elucidates the relative sensitivity of model performance to each parameter (Fig 3b-d). Model performance was most sensitive to $k_1$ for the parameter space we explored, with mean MAE values ranging from a minimum value of MAE = 0.0438 at $k_1 = 8.38 \times 10^{-6}$ min$^{-1}$ to MAE = 0.181 at $k_1 = 10^{-2}$ min$^{-1}$, reflecting the clustering of best-fit $k_1$ values (Fig. 3b). We note that $\gamma = 2.5$ was independently calibrated; it

is possible that if our analysis also explored a range of $\gamma$ values, then the range of acceptable $k_1$ may be broader. In contrast, mean MAE values are higher and less variable when binned by $k_2$ (MAE = 0.105-0.122) and $\varepsilon$ (MAE = 0.109-0.114), suggesting that annual model performance is less sensitive to variations in these parameters (Fig. 3b,c).

## 6 Discussion

### 6.1 Model performance

Our flow-history model for $\tau_c^*$ (Equations 1, 2) performs better than a constant entrainment threshold, as indicated by lower MAE between data and model (Fig. 3a). The model captures both first-order strengthening and weakening trends seen in the field data. These include progressive increases in $\tau_c^*$ from lower discharges (Fig. 3e), to sudden decreases in $\tau_c^*$ following a large flood early in the season (Fig. 3f), to a smaller decrease in $\tau_c^*$ following a late-season flood after the riverbed

may have had more time to strengthen (Fig. 3g), to intermittent but repeated weakening events across a season (Fig. 3h). When each year is calibrated separately ("annual" calibration), the model unsurprisingly performs better than when using the single set of parameters that minimizes MAE averaged across all years ("combined" calibration). Nonetheless, the "combined" best-fit parameters still outperformed a constant $\tau_c^*$ assumption in a majority of years – particularly those with seasonal strengthening trends.

Calibrated model performance varies most with $k_1$, which governs the efficacy of strengthening processes (Fig. 3b, Eq. 1). $\gamma$ also influences strengthening but was independently calibrated and held constant in our analysis ($\gamma$ = 2.5; Fig. *2B*). At the Erlenbach, sediment-transporting flood events only comprise about 2% of the discharge record (Masteller et al., 2019). Thus, weakening parameters $k_2$ and $\varepsilon$ can only influence $\tau_c^*$ evolution during this portion of the record, when $\tau^*/\tau_c^* > 1$ (Equation 1). More generally, the transport capacity ($\tau^*/\tau_c^*$) at which the model terms combine to transition from overall strengthening to weakening varies for different parameter combinations (Fig. 1). In years without large floods, the strengthening term of Equation 1 dominates for most of the year, resulting in steady increases in $\tau_c^*$, such as in 1988 when floods did not exceed $\tau^*/\tau_c^*$ = 1.64 (Fig. 3e). Therefore, best-fit models for 1988 were insensitive to $k_2$ and $\varepsilon$, resulting in 248 parameter combinations that minimized MAE (Fig. 3e).

Bedload is transported infrequently in gravel-bed rivers because transport thresholds are rarely exceeded. Much of bedload transport occurs during discharges that are often relatively close to bankfull, and when shear stresses only slightly to moderately exceed $\tau_c^*$ (e.g., Emmett and Wolman, 2001; Parker, 1978; Phillips and Jerolmack, 2016; Pretzlav et al., 2020; Whiting et al., 1999). Therefore weakening processes, which in our model only occur when $\tau^*/\tau_c^* \geq 1$, can only reduce $\tau_c^*$ for a limited fraction of the full discharge record. Thus, we may expect more generally that strengthening processes are dominant for most of the time in gravel riverbeds relative to weakening processes that may only occur during floods. However, despite strengthening being active for the vast majority of the discharge record, the approximately normal distribution of $\tau_c^*$ suggests that weakening processes must act more rapidly than strengthening processes to maintain values of $\tau_c^*$ intermediate between $\tau_{cmin}^*$ and $\tau_{cmax}^*$. Future work could further explore the consequences of this difference in the total time over which strengthening and weakening processes may occur and their resultant impact on the time-averaged state of the riverbed, as reflected by distributions of $\tau_c^*$. At the Erlenbach, the model is largely successful in matching annual strengthening trends (Fig.

3a, e-h) with a narrow distribution of best-fit $k_1$ values and a single, independently-calibrated value of $\gamma = 2.5$. This suggests that the physical processes that are encapsulated in the strengthening term of Equation 1 may lead to predictable changes in similar field settings (e.g. Church et al., 1998; Masteller & Finnegan, 2017; Ockelford & Haynes, 2013).

In contrast, calibrations of weakening parameters find much more variability in best-fit $\varepsilon$ and $k_2$, and MAE is higher for weakening years (Fig. 2a). Annual best-fit results span the full range of both parameters (Fig. 3c,d). High-magnitude floods that cause weakening are relatively rare and short-lived. It is possible that a dataset with many more discharge-driven weakening events could more narrowly constrain these variables. However, a simpler interpretation consistent with our analysis is that weakening is inherently less deterministic than strengthening, and therefore more difficult to predict accurately. Weakening events can be de-coupled from flow, for example, if hillslopes supply sediment to the channel during inter-event periods, leading to a reduction in $\tau_c^*$ at low discharge. Indeed, Masteller et al. (2019) identified a minimum discharge for inter-event strengthening, below which $\tau_c^*$ becomes uncorrelated with flow magnitude. This loss of correlation could reflect instances where supply effects introduce event-scale variability in $\tau_c^*$. While we focus on evaluating a discharge-driven model for $\tau_c^*$, it is not intended to fully address all factors influencing variability in $\tau_c^*$. The discrepancies between observed $\tau_c^*$ and model predictions may highlight conditions where sediment supply significantly alters bed mobility, outweighing the flow history effects that are addressed here.

Fig. 3e-h illustrates how feedback parameter $B$ controls how $\tau_c^*$ changes in response to a given shear stress. Low $\tau_c^*$ (such as at the starts of 1988 and 2010, and following the large 2007 flood) corresponds to high B, which increases the strengthening term and reduces the weakening term (Equations 1,2). As relatively smaller shear stresses lead to progressive strengthening, $\tau_c^*$ increases, B decreases, and less strengthening occurs for a given increment of Shields stress, resulting in a gradual rollover in the rate of strengthening through time (e.g., Fig. 3e). Model response (i.e., $\partial \tau_c^*/\partial t$) is also influenced by $\tau_c^*$ through changes in transport capacity. For example, under strengthening conditions, even if $\tau^*$ remains constant, increasing $\tau_c^*$ would cause a gradual decrease in $\tau^*/\tau_c^*$, slowing the rate of strengthening.

Thus, the "memory" in this model is represented by the value of $\tau_c^*$, which integrates the effects of the history of both flow conditions and channel bed conditions. Model memories tend to be asymmetric through time in the sense that floods large enough to cause significant weakening will rapidly reset the memory to lower $\tau_c^*$ values; strengthening can only occur

gradually as it requires the cumulative effects of lower discharges over time. Conceptually, these memory effects relate to $\tau_c^*$

being a state variable for gravel-bed channels (Johnson, 2016). For the Erlenbach, our results using the calibrated model demonstrate that knowing $\tau_c^*$ prior to a given flood improves the prediction of transport during that flood. We view this model as a step towards a more complete understanding of mountain river morphodynamics. Our calibrated $\tau_c^*$ equation should be useful for improved modeling of channel transport and evolution, and as a component of landscape evolution modeling. When high-resolution discharge data is available for field sites, incorporation of a flow-dependent $\tau_c^*$ may improve quantitative

predictions of transport in gravel-bed rivers, although calibration to local conditions is likely necessary.

**6.2 Model implementation beyond the Erlenbach**

The calibration that we have performed here leverages an extensive dataset of direct measurements of $\tau_c^*$. While similar datasets are available for a small subset of rivers (e.g. Turowski et al., 2011), most gravel bed rivers lack time series data of $\tau_c^*$. However, we find that the range of temporal variability observed in our calibration dataset is consistent with the

existing data compilations of $\tau_c^*$ across a range of slopes (Fig. 2a) suggesting that these data compilations may provide reasonable preliminary constraints of minimum and maximum bounds on $\tau_c^*$ in future applications of the model. Reach-averaged starting values of $\tau_c^*$ could be estimated based on bed grain size and bankfull geometry. Additional model calibration will vary depending on the intended application of the equation. Calibration of the model over approximately 30 transport events may be needed to reliably capture the expected variability of $\tau_c^*$. This assumes that $\tau_c^*$ are normally distributed, as

observed at the Erlenbach by Masteller et al. (2019). However, the commonly used minimum sample size of 30 to characterize normal distributions also assumes independent observations, which does not apply here. An alternative approach could be to calibrate the model based on the number of subsequent events over which $\tau_c^*$ remains correlated. Masteller et al. (2019) also found a loss of correlation between $\tau_c^*$ values after 10-13 transport events. This number of events may be sufficient to calibrate the model to capture the trajectory of $\tau_c^*$ over time. We recognize that requiring 10–30 measurements of $\tau_c^*$ may not always be

feasible. Future studies should assess the necessary level of calibration for different applications. Further research is needed to evaluate the potential variability of the $k_1$, $k_2$, and $\varepsilon$ parameters, especially their sensitivity to discharge distributions, grain size distributions, and bed slope. Controlled flume experiments could offer a systematic approach to investigating this variability.

**6.3 Sediment supply and nonlocal effects**

Perhaps the biggest mechanistic limitation of our model is that it only accounts for discharge controls on evolving thresholds, even though sediment supply has also been shown to explicitly influence transport rates in the Erlenbach data (Turowski et al., 2011, Rickenmann, 2020; 2024). In flumes, it is straightforward to impose the upstream sediment supply, measure the flux exiting the flume, and simultaneously measure changes along the flume bed (e.g., surface grain size distributions), allowing thresholds to be evaluated through time as a function of supply (e.g., Johnson, 2016). While it is possible to constrain temporal variations in upstream sediment supply in field settings (e.g., Hassan and Church, 2001; Rickenmann, 2020), these data are far more difficult to measure and less widely available than discharge timeseries for gravel-bed rivers. We are unaware of field monitoring sites that directly measure comparable timeseries of transport data in sequential channel reaches, making it difficult to directly isolate supply controls on threshold evolution in gravel bed rivers. Some sources of sediment supply into a given channel reach, such as shallow landslides a short distance upstream not triggered by recent precipitation, may be uncorrelated with channel discharge. However, the timing and magnitude of many processes that supply sediment to channels, such as bank failures, debris flows, and shallow landslides driven by recent precipitation, are likely correlated with timeseries of channel discharge (Turowski et al., 2013). In addition, sediment supplied from farther upstream in a watershed is transported into a given reach by channel flow. Seasonal trends in supply (such as from increased hillslope deposition during winter months), followed by subsequent snowmelt or storm flow that progressively transports the sediment (e.g., Moog and Whiting, 1998; Mao et al, 2014), may cause threshold evolution that correlates with cumulative seasonal discharge (e.g., Pretzlav et al., 2020). In other words, discharge and upstream sediment supply are not entirely independent over the timescales of threshold evolution due to floods. Given these correlations between sediment supply delivered to a channel reach from upstream and discharge, the local discharge timeseries may be able to implicitly account for some temporal variations in local supply, and therefore may be able to explain some supply-dependent $\tau_c^*$ variability. The degree of correlation between supply timeseries and discharge timeseries would likely vary among watersheds based on dominant processes. Future work should attempt to disentangle how sediment supply influences parameter calibrations.

Nonlocal controls on sediment transport will fundamentally limit how well local discharge timeseries can explain local sediment transport rate. Previous work demonstrates that bedload transport is a nonlocal process because the flux at a location within the channel reflects not only local conditions, but also spatial and temporal variations in the flow and sediment

flux from upstream (e.g., Foufoula-Georgiou and Stark, 2010; Furbish et al., 2017; Martin et al., 2012). Variability from
nonlocality limits the accuracy of all models for calculating bedload flux at a specific location based on local shear stress. This
challenge is not unique to our threshold evolution equation. Local flux models also cannot capture spatial and temporal grain
dispersion which is as important as advection for understanding bedload transport through river networks and responses to
perturbations (e.g., Bradley, 2017; Pretzlav et al, 2021; Fan et al., 2016). Our model could be applied to better determine the
extent to which threshold variability (and associated transport rate variability) is a deterministic function of discharge (as
Equation 1 attempts to represent), and how much of the local transport rate signal is stochastic variability, influenced by a
variety of interrelated factors including nonlocality and sediment supply. Future work could also explore how the "memory"
of past conditions at a given location, imperfectly encoded in $\tau_c^*$, depends on both local discharge variability and nonlocal
supply effects.

**7 Conclusion**

Our study presents a flow history-dependent model for critical Shields stress $\tau_c^*$ (Equations 1, 2). Calibrated using a
23-year record of river flow and the onset of sediment transport events, our model successfully captures observed trends in
$\tau_c^*$ evolution at our field site, including seasonal strengthening, rapid weakening following large floods, and gradual riverbed
recovery over time. While strengthening processes appear to be relatively predictable across different years, weakening
mechanisms exhibit greater variability, likely due to the stochastic nature of high-magnitude floods and external sediment
supply effects. Despite these uncertainties, our results demonstrate that accounting for flow-history effects by evolving $\tau_c^*$ has
the potential to significantly improve predictions of sediment mobility and bedload transport compared to a constant transport
threshold.

Building on the development and calibration of the model described in this contribution, the best-fit model parameter
values found in our study could be used as specific predictions to be independently tested using other field and flume data; we
do not yet know how consistent model parameters may be across gravel-bed channels. Model performance could also be
assessed at the scale of individual events using continuous bedload measurements, rather than just thresholds at the start of
events as done here. The model does not try to isolate granular interaction-based processes that likely cause strengthening and

weakening but rather lumps processes together using empirical parameters. Quantifying the systematics, inherent variability, and dominant processes involved in bed weakening warrants additional study. We suggest that a combination of discharge-based controls on $\tau_c^*$ (as explored here) and sediment-supply controls on $\tau_c^*$ (e.g., Recking, 2012; Johnson, 2016; Rickenmann, 2020; Rickenmann, 2024) may be able to explain much of the deterministic variability in threshold evolution and sediment transport rates in gravel-bed rivers.

**Code availability**

A working version of the code used to complete the model runs, associated best-fit model runs, and a summary of MAE values for all model runs produced during this research are publicly available through Zenodo (Masteller et al., 2025). Due to file upload limits of Zenodo, additional model runs are available by request to the corresponding author.

**Data availability**

The discharge and sediment transport time series data from the Erlenbach is publicly available at Rickenmann et al. (2020).

**Author contribution**

J. Johnson and C. Masteller formulated the concept of the study and the state function, with input from J.M. Turowski and D. Rickenmann. C. Masteller completed the model runs, calibration, and error analysis. Field data were collected and provided by D. Rickenmann and J.M. Turowski. The manuscript was written by C. Masteller with contributions from all authors.

**Competing interests**

Some authors are members of the editorial board of journal Earth Surface Dynamics.

**Acknowledgements**

Figure 1 utilizes perceptually uniform *scientific colourmaps* (Crameri et al, 2020). We greatly appreciate thoughtful reviews by Elowyn Yager and Shawn Chartrand.

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
