# Peer review of "Modeling memory in gravel-bed rivers: A flow history-dependent relation for evolving thresholds of motion"

_EGUsphere, 2024_

## Referee Comment (RC1)

The submitted paper develops a novel theory for the temporal evolution of the critical Shields stress as a function of flow magnitude. The theory is calibrated using field data to show that it improves estimates of critical Shields stresses in a gravel bed river. This is the first paper to develop an equation that predicts both flow strengthening and weakening effects on critical Shields stresses. It is also the first paper, to the best of my knowledge, that develops such a theory using field data; all other equations for flow effects on temporal changes in critical Shields stresses are based on laboratory data. I believe this study constitutes a major step forward in our potential to predict critical Shields stress changes over time. I have a few major comments that I think should be easy to address, which are mostly about clarifying the assumptions/calibration in the equations or placing this work in the context of equation application. Other than these comments and some minor line by line comments, I think the paper is ready for final publication. -Elowyn Yager

Major comments:

Calibration of $\tau^*c\_min$ and $\tau^*c\_max$: This may just be a matter of preference, but I think that Figure S1 (how $\tau^*c\_min/max$ is calibrated) might be better suited in the main text rather than in the supporting information because this demonstrates how the empirical parts of your main equations were developed? I think it might help your reader better understand, for example, why $\tau^*c\_max$ and $\tau^*c\_min$ vary with slope? Also since all assumed parameter values impact the equation accuracy and potential future application, I think it is useful for your reader to see in the main text how this calibration was performed? I think you also might need to discuss (in the main text) that you are calibrating $\tau^*c\_min/max$ to spatial (and not temporal) variations in observed $\tau^*c$. This method of calibration indirectly assumes that all measured spatial variability in $\tau^*c$ (for a given slope) is because of the same factors that would cause temporal changes in $\tau^*c$ in a given stream. However, variations of $\tau^*c$ between streams have been attributed to factors that may be independent of those that cause $\tau^*c$ temporal variations and I think this likely needs to be acknowledged? For example, this scatter in $\tau^*c$ between different streams with a given slope has been attributed to differences in relative roughness effects (e.g., Lamb et al., 2008, 2017) or differences in the underlying grain size distribution (e.g., Kirchner et al., 1990; Buffington et al., 2002; Shvidchenko et al., 2001). To help support your assumption of a kind of space for time substitution, could you plot the measured values of $\tau^*c$ for the Erlenbach on Figure S1? Or could you simply place arrows or lines on this plot that denote the range of temporal variation in $\tau^*c$ in the Erlenbach? If the range of temporal variations in $\tau^*c$ in the Erlenbach is similar to the spatial variations in $\tau^*c$ between many different sites, this could help further justify your assumption that all (or most) spatial variations in $\tau^*c$ are being caused by the same factors that control

temporal variations in tau*c at a given site (i.e., that you can use the range of spatial variations as a proxy for the range of temporal variations)?

Calibration of gamma: Similar to my comments on tau*c_min and tau*c_max, I think (again, maybe a matter of preference) the material on gamma calibration (Figure S2 and associated text) might be better suited to be placed in the main manuscript rather than as supporting material. Since the calibration of gamma affects all other parameter values and is a major part of the proposed equations, I think it likely belongs in the main text?  I think you have space (?) within the journal page limitations to move this text to the main document and Figures S1 and S2 could be combined into different panels of the same figure in the main text?  I also really appreciated the discussion about the uncertainties associated with the gamma calibration using the sand data/equation of Paphitis and Collins (2005).  I understand that grain size does not enter directly into their equation but it will enter into your conversion from utc to tau*c.  What grain size did you use in this conversion, some representative grain size from their experiments, the grain size from the Erlenbach, or some other size?  To further help make your case that using sand data to calibrate gamma is relevant for gravel bed channels, I wonder if you could also frame this analysis not only in terms of particle Reynolds number values but also tau*c values?  For example, are your back calculated values of tau*c (from utc from their equation) within the range of tau*c values typically reported for (hydraulically rough flow) gravel bed rivers?  If these tau*c are in the reported range for gravel rivers, I think this would further support your reasoning of applying this sand-based equation to gravel beds since all you really care about is tau*c and the time derivative of tau*c anyway?

Application of equation: The discussion mentions that the proposed equation could be used to improve critical Shields stress estimates at other field sites as long as some site specific calibration is conducted. I think it might be useful here to discuss what level of calibration would be needed for practical application of this approach?  You conducted extensive calibration of the equations because you had measured tau*c over time in the Erlenbach. In a river in which tau*c is not known and a user would instead like to predict tau*c, what type of calibration would be necessary?  Would someone need to measure tau*c over time for a certain period of time? Would calibration for one year of data on tau*c be sufficient or would many years of data be needed? If many years of data are needed, would this intensive data requirement potentially limit the application of this approach to very well studied rivers?  Or would only relatively limited data be needed for calibration, which would allow for a more broader application of this approach?  I think some discussion on this would be really helpful to understand how someone might apply your approach and the potential data-based limitations in using the approach?

Minor comments by line number/figure number etc.

36-39. "For example, hysteresis in bedload transport rates is often observed between the rising and falling limbs of individual floods (Hsu et al., 2011; Mao, 2018; Mao et al., 2014; Pretzlav et al., 2020; Reid et al., 1985; Roth et al., 2017). Dynamic threshold evolution over the duration of a flood event is implied by the observed change in bedload transport rate."I am not fully sure that you can state that all changes in bedload transport during an event are caused by threshold evolution? Changes in bedload transport rates during an event (hysteresis) could be caused by other factors. Transport rates could also change over an event because the flow hydraulics within a channel or the morphology have evolved during an event, which will both alter the applied shear stress without necessarily having to change the threshold of sediment motion. Similarly, many studies on hysteresis attribute changes in bedload transport rates to changes in sediment supply (e.g., landsliding) to the river during an event. Although sediment supply can influence the threshold of motion, it can also influence other variables that control the bed load transport rate such as bed grain size, channel bed roughness, channel topography etc. Hysteresis in bedload is likely caused by a variety of factors and I think you probably need to reword this text slightly?

50-54 and other locations. Thanks for this citation (!) but Yager et al., 2012 didn't investigate changes in the critical Shields stress after floods as implied here? They attributed increases in bedload transport rates after floods to a higher sediment supply during floods from landsliding and showed that this sediment supply would alter the applied flow shear stress by changing channel morphology? I think this might be a more relevant reference for when you are discussing sediment supply effects in the introduction and discussion rather than as cited in this location and others later on?

70-75. I think you might want to briefly discuss and review here the other developed equations for the temporal evolution of critical shear stresses as a function of flow properties? I think this could more specifically set the stage for missing component of these equations that you are explicitly trying to address here (e.g., none calibrated with field data, none include both strengthening and weakening)? I think this could better highlight the novelty of what you have done. For example, you could mention Ockelford et al. (2019) used flume experiments to develop an empirical equation for the temporal strengthening of the critical shear stress using the duration of a certain flow magnitude. Also, a brief discussion of Paphitis and Collins (2005), and their flume based equation for strengthening based on flow conditions, could also be mentioned here?

Equation (2) and lines 99-100. I understand how B=0 when tau*c=tau*c_max but I don't understand under what conditions B is equal to one because I think (?) the equation becomes undefined when tau*c=tau*c_min? Can you please explain under what

conditions B=1? Also, can you explain how you deal with the situation when tau*c=tau*c_min?

Figure S1 caption. Can you explain how field and flume data are equally weighted when conducting the regression? How are you assigning weights to the data to offset the greater number of datapoints in flume experiments? Are the labels of the figure correct because the caption says there are 3.5 more datapoints from flume experiments but in the figure, it looks like there are more field data (blue diamonds) than flume data (red circles)? The figure caption also says the best fit exponent is 0.36 but in the figure legend, the exponent appears to be 0.7 in all equations; can you please make these consistent?

170-200. In the results, a mix of median parameter values (k1, k2 etc.) for the annual best fits and mean variable values for the average best fit are used. Is there a reason for mixing the use of medians and means of parameter values? I think it might be better to use one of these consistently (median or mean parameter value) or explain why you are using medians for the annual best fits instead of means, which would be more directly comparable to the average best fit parameter values for all years combined?

203-205. But is the lower range of values for k2 and epsilon because the model is less sensitive to these parameters or could it also be because you only had three years in which weakening (which these parameters represent) was dominant, which I think you kind of imply in the discussion? Can you please clarify here?

---

## Referee Comment (RC2)

**Reviewed manuscript**: Modeling memory in gravel-bed rivers: A flow history-dependent relation for evolving thresholds of motion

**Manuscript Authors**: C. Masteller and colleagues

**Journal**: Earth Surface Dynamics

**Reviewer**: Shawn M. Chartrand

Summary and Contribution:

The authors present an empirical model which describes the discrete time evolution of the critical dimensionless Shields number (similarity-based threshold for sediment transport; note that I use "number", "condition" and "stress" below in an interchangeable way) using a unique field data set of streamflow discharge and sediment transport with the latter measured using an impact plate system installed within the Erlenbach located in the Swiss Prealps. The data are unique because sediment transport is quantified directly over annual hydrographs (along with observations of streamflow) with the impact plate system rather than using wading-type measurements combined with rating curves or transport functions for times lacking observations through measurement. As a result, the authors data permit them to identify streamflow conditions associated with the onset of bedload transport, removing substantial uncertainty in estimation of the critical dimensionless Shields number. The empirical model builds from prior work (Johnson, J.P.L., *Earth Surface Dynamics*, 2016) with revisions to the original model focused on describing the discrete time dependent behavior of the critical dimensionless Shields number in terms of so called "strengthening" and "weakening" processes. The model faithfully (and impressively) captures annual trends to the critical dimensionless Shields stress based on the available records during years when "strengthening" dominates and/or "weakening" events do not occur, with more variable performance for years when "weakening" dominates. To my understanding the authors model is the first of its kind in the geomorphic literature, and I congratulate them on their excellent work.

Based on several readings of the manuscript, I believe some effort is needed to address the comments and suggestions I raise below. I do not have any serious criticisms of the authors work, however, there are some conceptual points (bigger picture comments) that would benefit from thought and discussion in the manuscript. I think the manuscript is well written, and the figures nicely developed such that they add important illustrative dimensions and information to the narrative of the text. I appreciate the authors inclusion of discussion elements which add some critical review of the proposed empirical model making it clear that they have thought about their approach and results from a variety of viewpoints. Some specific examples of the points raised below include consideration of bedload transport in rivers as a "nonlocal" process which in part helps set the context of the measurement system as well as the fact that the authors calculation approach mixes time and space considerations, and additional discussion focused on the nuances around the interdependence of (in-channel) sediment supply and the critical dimensionless Shields condition. I also have a few questions around the parameter *Beta* that would benefit from clarification, most importantly, I cannot recover a (0,1) behavior of *Beta* with the form of Eq. 2 presented in the manuscript (I get negative values).

I want to thank the journal editors and authors for the opportunity to review the submitted manuscript, and for your patience as I completed my review. I hope that my comments are helpful to the authors.

**Bigger Picture Comments (in no particular order)**

1.  **Transport as a nonlocal process:** There has been important work completed that illustrates how bedload transport in rivers is a non-local process, i.e. that the transport (activity in the case of the authors work) of particles close to the bed surface as measured at any particular point along a river profile *x* and at some time *t* is a function of transport processes upstream of *x* (within some finite distance set by the *pdf* of particle travel distances) and for some finite time interval prior and leading up to *t* (within some finite time interval set by the *pdf* of particle travel times; see Furbish et al., *GBR*, 2017 for a clear discussion; see Foufoula-Georgiou and Stark, *JGR*, 2010 for a broader discussion). The implications of bedload transport as a nonlocal process has relevance to the authors work in a number of ways: (1) it provides context for interpreting the time series of particle impacts at the Erlenbach measurement site [and hence time dependency of the dimensionless Shields stress and the critical dimensionless Shields stress] by offering a conceptually useful way to understand the authors "noisy transport data" as more of an expected outcome based on the explicit dependence of transport on time and space; (2) it offers a broader perspective to conclusions reached regarding deterministic variability of threshold evolution; (3) it elaborates the context for the authors concept of "memory" in that nonlocality provides a more concrete way to frame the authors proposal that memory "integrates the effects of the past history of both flow conditions and channel bed conditions" (lines 253-254, page 12 of the manuscript; although I also recognize that the authors idea of memory involves much longer time scales); and (4) it helps to better frame the authors calibration and application of Eq. 1 to the Erlenbach data because the authors approach mixes space and time effects, and a nonlocal perspective naturally reflects these two aspects of transport processes. In summary, the authors relate time variations of the critical dimensionless Shields number to conditions upstream of the point of observation. A nonlocal perspective of bedload transport can help the authors, to some degree, make these points in more concrete ways.

2.  **Equation 1 and *Beta*:** In section two the authors discuss their model for the time evolution of the critical dimensionless Shields condition. I think this section will benefit from more direct discussion of the development part of the model. As it stands the section explains the components of the model and why specific elements, etc. are justified, but their discussion does not provide much detail on the model development side. For example, if someone else attempted to recreate the authors model working from the existing text and supplemental information alone, do they have enough information to guide their thinking and decision making to eventually lead them to the form of the model presented in Eq. 1? Basically, what were the key steps or decisions made that led the authors to the present model formulation. To be clear, I am not suggesting that the authors list out every decision made along the way to the formulation of Eq. 1, but rather they provide enough information to better understand the key steps in getting there. Perhaps this can be addressed by prefacing the presentation of Eq. 1 by stating explicit hypotheses or specific ideas that underpin the authors thinking (this is briefly done at the end of section one but I am suggesting it is done in relation to presenting the proposed empirical model). Among other reasons, providing more clarity around model development will help future readers as they attempt to apply and test the model to any particular circumstance. With this in mind, it may be helpful to write out the discretized form of Eq. 1, or specify the time marching components of Eq. 1 with notation so it is clear how the initial and time dependent values are specified in the calculation procedure.

    The present text lists *Beta* (Eq. 2) as a ratio of the differences between the max/min critical dimensionless Shields stress and the time dependent critical dimensionless Shields stress,

respectively. Based on my reading of the manuscript I have assumed that the authors used max and min values of 0.36 and 0.036, respectively, for all associated calculations (Fig. 1 caption). However, I am not sure if this is the case. If this is true and using the form of *Beta* given in Eq. 2, it seems that *Beta* should be negative at times when the critical dimensionless stress > the min dimensionless critical Shields condition (0.036?), and < the max dimensionless critical Shields condition (0.36?)-- for example if the critical dimensionless Shields number has a value of 0.05. However, the authors state that *Beta* ranges from a value of 0-1 (lines 99-100). What am I missing? What were the values of the max and min dimensionless critical Shields stress used in the calculations of Eqs. 1 and 2? Did they change in time? Why is the authors form of *Beta* different from the form given by Johnson, 2016 (Johnson, J.P.L., Earth Surface Dynamics, 2016)? Were the max and min values calculated with the power law forms given after Eq. 2? If so, what was the value(s) of *S* used (based on the min/max values given in the Fig. 1 caption, the power law forms of the min/max suggest *S* ~ 0.11)? Also, based on the form of Eq. 2, it is difficult for me to imagine how *Beta* takes a value of 1 because the max and min values of the critical dimensionless Shields number by definition will never be equal. Clarification around *Beta* will be helpful.

Last, the supplemental information provides important information related to calibration of Eq. 1. based in part on the work of Paphitis and Collins, 2005 (Paphitis and Collins, *Sedimentology*, 2005). It is important that this information is presented in the main text because it is key to calibration of the *Gamma* exponent of Eq. 1 term 1, and second because the referenced work was conducted experimentally using sand sized particles, which diverges from the Erlenbach field conditions.

3. **Critical Shields condition:** Conceptually, I stumble over how to disentangle the critical dimensionless Shields condition, the dimensionless Shields condition and the sediment supply. The critical dimensionless Shields condition and the dimensionless Shields condition are derived quantities that are calculated based on specific information (e.g. the authors estimation of the dimensionless Shields number and the critical dimensionless Shields number relies on an empirical rating curve relating streamflow discharge to the local average dimensional shear stress [Yager, E.M., PhD thesis, 2006]). Meaning, it is not possible to directly measure a critical dimensionless stress or Shields condition. The authors record of particle impacts removes a substantial degree of uncertainty related to when transport begins in the monitored section of the Erlenbach. However, the principal metrics are still subject to calculation. On the other hand, sediment supply delivered to some position *x* and for some time interval *t+Δt* is a tangible thing, something that given adequate technology, etc. can be measured and quasi-directly quantified (or at least approximately so) because it is a physical response. In rivers, for example, the diligent and careful use of sediment baskets and similar passive measurement apparatuses situated in the streambed can provide a reasonable record of supply magnitude, grain size composition and the rough time interval over which a basket fills (e.g. Hassan and Church, *Water Resources Research*, 2001). At several locations in the manuscript the authors state or use other studies to suggest that sediment supply influences threshold evolution (lines 43-44), and in turn that supply depends on thresholds (lines 58-60). I understand conceptually how thresholds and sediment supply are inter-related, noting that "sediment supply" can mean a couple different things—i.e. in-channel storage, hillslope derived, and so forth. It would be helpful if the authors developed an expanded discussion of sediment supply vs. thresholds in which they more carefully step through the nuances of how these are inter-related and inter-dependent, and what is explicitly meant by sediment supply in the context of the manuscript. The last paragraph of section six could be

suitable to expand the discussion, although it would be more impactful to have this presented in section one as the reader will have a clearer picture when reading the remainder of the manuscript.

4. **Bedload transport and threshold conditions:** There are discussion elements which frame bedload transport around threshold conditions, inter-connections between these conditions and gravel-bed river geometry (e.g. lines 22-23; lines 222-224) and the nature of threshold conditions during floods or transporting events (e.g. lines 22-23, lines 38-39, lines58-61). In several cases there is important literature missing from the discussion. For example, ideas around bedload transport, transporting floods and gravel-bed river geometry have been the subject of significant field-based data collection efforts, some of which provide results not as definitive as that suggested in the present form of the manuscript. Whiting et al., 1999 (Whiting et al., *GSAB*, 1999) present a comprehensive dataset based on hundreds of bedload measurements across more than 10 headwater rivers which suggests that gravel-bed river geometry is maintained in the Idaho batholith (a snowmelt dominated system) at flows less then bankfull (~0.80 bankfull), with the common 1- to 2-orders of magnitude variability in the sediment-flow rating curves. There are many other examples in the literature as well (that I know the authors are aware) which suggest that bankfull flow is not necessarily the most important flow in maintaining channel form (geometry). I raise this point because the subject is the matter of significant debate in fluvial geomorphology, and presenting a more balanced picture of what the literature suggests seems appropriate.

Minor and Editorial Comments

**Lines 16-17 (comment also relates to lines 237-238).** The sequence, timing and magnitude of significant precipitation events and heatwaves (both of which cause floods) are stochastic. Because "weakening" events are directly linked with floods, the authors conclusion that weakening events are more stochastic than strengthening ones is clear. Use of the phrase "...suggests that flood-induced bed weakening is more stochastic and less predictable then strengthening." is a little confusing. Are there additional mechanisms not related to flood events (this would include mass movements, etc.) that could cause weakening? I guess I am unsure whether this is a surprise or unexpected from the authors point of view? I think the authors specifying "more stochastic and less predictable" is what is causing my confusion. Also, what does "more stochastic" mean?

**Lines 22-23**. What do "close" and "floods" mean? Can the authors be more specific?

**Lines 23-26**. Blom et al., 2017 conclude that the influence of climate change (and hence extreme floods) to river geometry in the zone downstream of the hydrograph boundary layer and upstream of the terminal backwater zone may be negligible (page 19 of Blom et al., 2017). How does this fit into the concept of mapping "climate onto fluvial processes"?

**Lines 27-29**. I had to read this sentence a few times to understand it. Can the authors re-phrase for clarity?

**Lines 57. Minor point:** Do all sediment pulses cause disequilibrium? Sediment pulses are commonly of short relative timescales. Ideas around disequilibrium can be associated with longer relative timescales. Pulses can disrupt local bed elevations, grain size populations and hence local transport rates—this fits at least two ideas of disequilibrium. But pulses can also fall under the concept of dynamic equilibrium. I am

wondering if/how disequilibrium as used in this sentence differs or is similar to the use of the same word in the following sentence?

**Lines 58-70.** I think the work of Moog and Whiting, 1998 (the authors include this work in their references) is relevant to the discussion here. The key from Moog and Whiting relates to hysteresis and their data which suggest that prior to the occurrence of the estimated transporting flow each season, there was higher bedload transport for a given flow than afterward. This trend was attributed to limitation or exhaustion of in-channel sediment supply as the snowmelt hydrograph progressed. This comment in part relates to my "bigger picture" comment above related to sediment supply and the Shields condition.

**Lines 165-166.** How do you know that certain data are "outliers"? Does the concept of an outlier make sense in an inherently "noisy" system? Even time series of flux or transport in the most simple of experiments is, for example, noisy (for example, see Fig. 6 of Ancey et al., *Physical Review E*, 2006).

**Lines 179-186 – Figure 2.** Nice figure with lots of great information. What do the black dots in the lower panels represent? I read carefully and could not find mention of what they represent.

**Lines 219.** "...higher *relative* transport capacities"?

**Lines 228.** There is a missing word towards the end of the line.

**Lines 257-258.** The sequence, timing and magnitude of future floods is a stochastic phenomenon dependent on future climate conditions, etc. I believe it is generally held that future conditions can be "projected" (not predicted) when there is a dependence on climate-related phenomena because of the probabilistic nature of the problem. Perhaps I misunderstand the intent of the sentence?

**Lines 258-262.** I encourage the authors to provide a more comprehensive discussion of how their model of the critical dimensionless Shields stress can be applied in other circumstances, with particular details related to what data, at a minimum, are necessary to locally calibrate their Eq. 1. I think data additional to a high-resolution discharge time series is necessary.

**Lines 263-265.** I don't understand how "transport disequilibrium" influences transport rates? My understanding of the idea is that transport disequilibrium depends on how observed transport compares to calculated transport (Rickenmann, D., *Water Resources Research*, 2020).

**References not in the manuscript**:

Ancey, C., Böhm, T., Jodeau, M., and Frey, P.: Statistical description of sediment transport experiments, Physical Review E, 74, 11302–11302, https://doi.org/10.1103/PhysRevE.74.011302, 2006.

Foufoula-Georgiou, E., and C. Stark (2010), Introduction to special section on Stochastic Transport and Emergent Scaling on Earth's Surface: Rethinking geomorphic transport—Stochastic theories, broad scales of motion and nonlocality, *J. Geophys. Res.*, 115, F00A01, doi:10.1029/2010JF001661.

Furbish, D. J., Fathel, S. L., and Schmeeckle, M. L.: Particle Motions and Bedload Theory: The Entrainment Forms of the Flux and the Exner Equation, in: Gravel-bed rivers, Wiley-Blackwell, 97–120, https://doi.org/10.1002/9781118971437.ch4, 2017.

Hassan, M. A., and M. Church: Sensitivity of bed load transport in Harris Creek: Seasonal and spatial variation over a cobble-gravel bar, *Water Resour. Res.*, 37(3), 813–825, doi:10.1029/2000WR900346, 2001.

---

## Author Comment (AC1)

The submitted paper develops a novel theory for the temporal evolution of the critical Shields stress as a function of flow magnitude. The theory is calibrated using field data to show that it improves estimates of critical Shields stresses in a gravel bed river. This is the first paper to develop an equation that predicts both flow strengthening and weakening effects on critical Shields stresses. It is also the first paper, to the best of my knowledge, that develops such a theory using field data; all other equations for flow effects on temporal changes in critical Shields stresses are based on laboratory data. I believe this study constitutes a major step forward in our potential to predict critical Shields stress changes over time. I have a few major comments that I think should be easy to address, which are mostly about clarifying the assumptions/calibration in the equations or placing this work in the context of equation application. Other than these comments and some minor line by line comments, I think the paper is ready for final publication. - Elowyn Yager

We thank the reviewer for her thoughtful and thorough review of our submission. Below, we respond to the reviewer's comments and describe the edits we plan to make to the text addressing their helpful review.

Major comments:
Calibration of tau*c_min and tau*c_max: This may just be a matter of preference, but I think that Figure S1 (how tau*c_min/max is calibrated) might be better suited in the main text rather than in the supporting information because this demonstrates how the empirical parts of your main equations were developed? I think it might help your reader better understand, for example, why tau*c_max and tau*c_min vary with slope?

We agree with the reviewer's suggestion. We have refined Figure S1 and plan to include it in the updated version of our MS as a new Figure 2. We have also added the observed Erlenbach values of t*c as the reviewer later suggests. We also added supporting language to the text to clarify how this parameterization was developed.

Here is the revised figure and new caption:

[Figure]

Figure 2. a) Compiled $\tau_c^*$ data as a function of slope from both field studies (grey diamonds) and flume experiments (open circles). Range of $\tau\_c^*$ from the Erlenbach field site is plotted in red. Compiled data are limited to slopes, S < 0.2 and median grain sizes, D50 ≥ 2 mm. Most data were compiled by Prancevic and Lamb (2015), building on Buffington and Montgomery (1997), with additional data from Olinde (2015) and Lenzi et al. (2006). b) . Calibration of strengthening term exponent, γ, based on Paphitis and Collins (2005) for both the shortest conditioning time (5 minutes) and longest conditioning time (120 minutes) spanned by the Paphitis and Collins (2005) data. Regressions to their best-fit empirical equation give gamma exponents within uncertainty of each other.

We will also revise the text, starting at Line 95 to read:

> Figure 2a shows that the empirical $\tau\_cmin^*$ and $\tau\_cmax^*$ relations in Equation 2 capture the slope-dependence of gravel thresholds of motion compiled in both flume and field settings, while asymptoting to reasonable bounds at low channel slopes (e.g. Johnson, 2016; Lamb et al., 2008; Prancevic & Lamb, 2015). Field and flume data were weighted equally in the best-fit regression, removing possible bias from there being ~3.5 times more flume data points. Minimum and maximum bounds were determined visually to accommodate almost all data points, assuming a consistent best-fit exponent (0.7). Our model is similar in form to that of Johnson (2016)."

Also since all assumed parameter values impact the equation accuracy and potential future application, I think it is useful for your reader to see in the main text how this calibration was performed? I think you also might need to discuss (in the main text) that you are calibrating tau*c_min/max to spatial (and not temporal) variations in observed tau*c. This method of calibration indirectly assumes that all measured spatial variability in tau*c (for a given slope) is because of the same factors that would cause temporal changes in tau*c in a given stream. However, variations of tau*c between streams have been attributed to factors that may be independent of those that cause tau*c temporal variations and I think this likely needs to be acknowledged? For example, this scatter in tau*c between different streams with a given slope has been attributed to differences in relative roughness effects (e.g., Lamb et al., 2008,

2017) or differences in the underlying grain size distribution (e.g., Kirchner et al., 1990; Buffington et al., 2002; Shvidchenko et al., 2001). To help support your assumption of a kind of space for time substitution, could you plot the measured values of tau*c for the Erlenbach on Figure S1? Or could you simply place arrows or lines on this plot that denote the range of temporal variation in tau*c in the Erlenbach? If the range of temporal variations in tau*c in the Erlenbach is similar to the spatial variations in tau*c between many different sites, this could help further justify your assumption that all (or most) spatial variations in tau*c are being caused by the same factors that control temporal variations in tau*c at a given site (i.e., that you can use the range of spatial variations as a proxy for the range of temporal variations)?

The reviewer is correct that this approach implicitly assumes that the range of variability in tau*c observed at a given slope is comparable to the variability in threshold introduced by flow history effects. However, the full range of tau*c observed at the Erlenbach is equivalent to the range of thresholds across the flume and field data compiled in Fig. 1A. Adding the Erlenbach range to Fig. 1A shows this nicely. We added the following to the main text to point this out and clarify:

Added at Line 100:
> We acknowledge that many other factors beyond slope may influence the possible range of $\tau\_c^*$ variability a given sites, including grain size distributions (e.g., Parker, 1990) and differences in relative roughness. In the absence of independent $\tau\_{cmin}^*$ and $\tau\_{cmax}^*$ constraints that could be used to describe a particular field site, the compilation should reasonably represent the range of possible values.

Added at (now) Line 617:
> The temporal variability in $\tau_c^*$ observed at the Erlenbach is equivalent to the full range of $\tau_c^*$ observed in flume and field data for equivalent slopes (Fig. 1A).

**Calibration of gamma:** Similar to my comments on tau*c_min and tau*c_max, I think (again, maybe a matter of preference) the material on gamma calibration (Figure S2 and associated text) might be better suited to be placed in the main manuscript rather than as supporting material. Since the calibration of gamma affects all other parameter values and is a major part of the proposed equations, I think it likely belongs in the main text? I think you have space (?) within the journal page limitations to move this text to the main document and Figures S1 and S2 could be combined into different panels of the same figure in the main text? I also really appreciated the discussion about the uncertainties associated with the gamma calibration using the sand data/equation of Paphitis and Collins (2005). I understand that grain size does not enter directly into their equation but it will enter into your conversion from utc to tau*c. What grain size did you use in this conversion, some representative grain size from their experiments, the grain size from the Erlenbach, or some other size? To further help make your case that using sand data to calibrate gamma is relevant for gravel bed channels, I wonder if you could also frame this analysis not only in terms of particle Reynolds

number values but also tau*c values? For example, are your back calculated values of tau*c (from utc from their equation) within the range of tau*c values typically reported for (hydraulically rough flow) gravel bed rivers? If these tau*c are in the reported range for gravel rivers, I think this would further support your reasoning of applying this sand-based equation to gravel beds since all you really care about is tau*c and the time derivative of tau*c anyway?

We have taken the reviewer's suggestion and combined Figures S1 and S2 into a new figure, now displayed as Figure 2 in the revised text. We have also added the details of the calibration of gamma to the main text as suggested by the reviewer, starting at Line 150.

To answer the reviewer's question regarding grain size: We are using reported values of critical shear velocity before and after low flow conditions reported by Paphitis and Collins (2005) for their coarse sand experiments (D=0.774 mm). Mathematically the grain size does not actually come into this calculation or conversion, because their equation is nondimensional, giving the ratio of critical shear velocity after flow conditioning to the critical shear velocity without flow conditioning. To convert from a shear velocity ratio to a shear stress ratio we only had to square the shear velocity ratio. While their results may depend on the grain size they used, the conversion from their nondimensional relation to ours does not. We tried to clarify this in the incorporation of the calibration into the main text.

Revised text starting at Line 150:

[revised manuscript text omitted]

Application of equation: The discussion mentions that the proposed equation could be used to improve critical Shields stress estimates at other field sites as long as some site specific calibration is conducted. I think it might be useful here to discuss what level of calibration would be needed for practical application of this approach? You conducted extensive calibration of the equations because you had measured tau*c over time in the Erlenbach. In a river in which tau*c is not known and a user would instead like to predict tau*c, what type of calibration would be necessary? Would someone need to measure tau*c over time for a certain period of time? Would calibration for one year of data on tau*c be sufficient or would many years of data be needed? If many years of data are needed, would this intensive data requirement potentially limit the application of this approach to very well studied rivers? Or would only relatively limited data be needed for calibration, which would allow for a more broader application of this approach? I think some discussion on this would be really helpful to understand how someone might apply your approach and the potential data-based limitations in using the approach?

Thank you for the thoughtful comment. Calibration requirements will vary depending on the intended application of the equation. The key question is whether the user wants to estimate the long-term variability in critical Shields stress, possibly by generating a probability distribution over a long discharge record, or if they want to refine estimates of bedload transport for individual events. Each application would require different calibration data based on its scope and required accuracy. While we cannot outline every hypothetical case in the main text, we have added a paragraph acknowledging potential calibration requirements and the need for further study on the sensitivity of model parameters to individual sites. We decided to focus on a similar approach to the analysis that we present in the paper and leave allusion of event-scale parameterization to the final paragraph.

If we want to reliably estimate the distribution of tau*c, a useful starting point is the "rule of thumb" from central limit theorem that approximately 30 independent samples are typically needed to capture the distribution of normally distributed data. Given that the Erlenbach tau*c data appear to follow a normal distribution, the model may require calibration based on at least 30 events to encompass the expected range of variability at a site. However, this assumes independence between measurements, which we know is not strictly true. Another approach could be to calibrate the model based on the timescale over which tau*c remains correlated. Masteller et al. (2019) found that at the Erlenbach, tau*c exhibits "memory loss" after approximately 10 events, suggesting that this number of events could be sufficient to calibrate the model to capture the trajectory of tau*c over time. However, a major challenge in model calibration will always stem from hydrologic variability and river self-organization itself. Small floods that strengthen the bed occur frequently, whereas weakening events which exceed the bankfull condition are rarer. As a result, robust calibration of weakening parameters will be significantly more difficult than for strengthening. Again, these approaches would be most applicable for studies focusing on only the initiation of motion, as done in our paper. We acknowledge that the requirement of 10-30 observations only to calibrate the model may be labor intensive, so more work is needed to better determine the potential variability in the k1, k2, and epsilon parameters introduced in the model. However, the variability of some of these parameters with flow history, bed slope, and grain size distributions can be more readily explored in flume experiments, as we suggested in the closing paragraph.

We've added the following paragraph to the discussion to clarify these points:

*The calibration that we have performed here leverages an extensive dataset of direct measurements of $\tau_c^*$. While similar datasets are available for a small subset of rivers (cites), most gravel bed rivers lack time series data of $\tau_c^*$. However, we find that the range of temporal variability observed in our calibration dataset is consistent with the existing data compilations of $\tau_c^*$ across a range of slopes (Fig. 2a) suggesting that these data compilations may provide reasonable preliminary constraints of minimum and maximum bounds on $\tau_c^*$ in future applications of the model. Reach-averaged starting values of $\tau_c^*$ could be estimated based on bed grain size and bankfull geometry. Additional model calibration will vary depending on the*

*intended application of the equation. To characterize long-term variability in $\tau_c^*$, calibration of the model over approximately 30 transport events may be needed to reliably capture the expected variability of $\tau_c^*$, assuming a normal distribution, as observed at the Erlenbach by Masteller et al. (2019). However, this commonly used minimum sample size assumes independent observations, which does not apply here. An alternative approach is to calibrate the model based on the number of subsequent events over which $\tau_c^*$ remains correlated. Masteller et al. (2019) also found a loss of correlation between $\tau_c^*$ values after 10-13 transport events. This number of events may be sufficient to calibrate the model to capture the trajectory of $\tau_c^*$ over time. We recognize that requiring 10–30 observations may not always be feasible, and future studies should assess the necessary level of calibration for different applications. Further research is needed to evaluate the potential variability of the k1, k2, and $\epsilon$ parameters, especially their sensitivity to discharge distributions, riverbed grain size, and bed slope. Controlled flume experiments could offer a systematic approach to investigating this variability.*

Minor comments by line number/figure number etc.

36-39. "For example, hysteresis in bedload transport rates is often observed between the rising and falling limbs of individual floods (Hsu et al., 2011; Mao, 2018; Mao et al., 2014; Pretzlav et al., 2020; Reid et al., 1985; Roth et al., 2017). Dynamic threshold evolution over the duration of a flood event is implied by the observed change in bedload transport rate."I am not fully sure that you can state that all changes in bedload transport during an event are caused by threshold evolution? Changes in bedload transport rates during an event (hysteresis) could be caused by other factors. Transport rates could also change over an event because the flow hydraulics within a channel or the morphology have evolved during an event, which will both alter the applied shear stress without necessarily having to change the threshold of sediment motion. Similarly, many studies on hysteresis attribute changes in bedload transport rates to changes in sediment supply (e.g., landsliding) to the river during an event. Although sediment supply can influence the threshold of motion, it can also influence other variables that control the bed load transport rate such as bed grain size, channel bed roughness, channel topography etc. Hysteresis in bedload is likely caused by a variety of factors and I think you probably need to reword this text slightly?

Fair point – reworded to: "Dynamic threshold evolution over the duration of a flood event **is one mechanism** that can results in the observed changes in bedload transport rate".

50-54 and other locations. Thanks for this citation (!) but Yager et al., 2012 didn't investigate changes in the critical Shields stress after floods as implied here? They attributed increases in bedload transport rates after floods to a higher sediment supply during floods from landsliding and showed that this sediment supply would alter the applied flow shear stress by changing channel morphology? I think this might be a more relevant reference for when you are discussing sediment supply effects in the introduction and discussion rather than as cited in this location and others later on?

Fair enough. In our reading of the paper, we interpreted the rapid reduction in sediment availability following an extreme event and the subsequent slowdown in in the changes in sediment availability (Fig. 1c) to imply a decrease in transport efficiency with time since the extreme event as the bed went from a relatively weak state to a relatively strong state. We removed the reference in cases where it was mis-referenced.

70-75. I think you might want to briefly discuss and review here the other developed equations for the temporal evolution of critical shear stresses as a function of flow properties? I think this could more specifically set the stage for missing component of these equations that you are explicitly trying to address here (e.g., none calibrated with field data, none include both strengthening and weakening)? I think this could better highlight the novelty of what you have done. For example, you could mention Ockelford et al. (2019) used flume experiments to develop an empirical equation for the temporal strengthening of the critical shear stress using the duration of a certain flow magnitude. Also, a brief discussion of Paphitis and Collins (2005), and their flume based equation for strengthening based on flow conditions, could also be mentioned here?

Thanks for the suggestion! We added a brief description of these models in the final paragraph of the introduction. It now reads:

> *While empirical evidence exists for systematic, flow strength-dependent temporal variations in thresholds for motion, few equations exist that quantify feedbacks leading to threshold evolution which can be incorporated into existing bedload transport models. Relatively simple model formulations have been proposed to describe temporal bed strengthening as a function of the duration of bed exposure to a constant, inter-event flow magnitude and an initial $\tau_c^*$ based on experimental data (e.g. Ockelford et al., 2019; Paphitis and Collins, 2005). However, because these models only focus on inter-event strengthening effects, they cannot capture decreases in $\tau_c^*$. Johnson's (2016) model predicts $\tau_c^*$ evolution as a function of sediment supply and allows for both strengthening and weakening effects. Nonetheless, this model is an incomplete*

*description of $\tau_c^*$ evolution because it does not account for riverbed strengthening or weakening directly caused by the flow. Notably, to our knowledge, none of these equations have been used to describe field observations of temporally varying $\tau_c^*$. Our goals in the present work are (i) to propose a new model in which $\tau_c^*$ evolves as a function of flow magnitude and encapsulates some memory of past shear stresses as reflected in the changing state of the riverbed, and (ii) to evaluate whether the model can broadly capture annual strengthening and weakening trends as a function of discharge, as observed in Erlenbach field data (Masteller et al., 2019).*

**Equation (2) and lines 99-100.** I understand how B=0 when tau*c=tau*c_max but I don't understand under what conditions B is equal to one because I think (?) the equation becomes undefined when tau*c=tau*c_min? Can you please explain under what conditions B=1? Also, can you explain how you deal with the situation when tau*c=tau*c_min?

We apologize. Our equation had a significant typo which we overlooked, and have now corrected. The equation now correctly gives 1 at tau*=tau*min, and 0 at tau*=tau*max.

Tau*c_max and tau*c_min represent bounds of our model. Because of the feedback term, B, tau*c can approach, but cannot reach these values. Equation 1 is only defined within these limits. We have modified the text to clarify this starting at Line 95:

> *In this treatment, we assume that $\tau_{cmin}^*$ and $\tau_{cmax}^*$ are limits that the threshold can approach but does not reach due to the feedbacks implemented by the B parameter (Equation 2). Equation 1 is only defined between these bounds.*

**Figure S1 caption.** Can you explain how field and flume data are equally weighted when conducting the regression? How are you assigning weights to the data to offset the greater number of datapoints in flume experiments? Are the labels of the figure correct because the caption says there are 3.5 more datapoints from flume experiments but in the figure, it looks like there are more field data (blue diamonds) than flume data (red circles)? The figure caption also says the best fit exponent is 0.36 but in the figure legend, the exponent appears to be 0.7 in all equations; can you please make these consistent?

Thanks for spotting the error in our figure legend. The labels were flipped. We have updated them in our revised figure. We have also corrected the best-fit exponent to 0.7 in both the figure and the text.

The regression was done using nonlinear curve fitting in Matlab, with all the field-derived data points together having equal weight in the regression as all the flume-derived data points. In other words, because there were ~3.5 times more flume data points than field data points, each field data point was weighted 3.5x more strongly in the regression compared to each flume point. Overall, the field and flume points each contribute half of the weight in the regression. We have added the following to the text starting at Line 100 to help clarify:

> *Field and flume data were weighted equally in the nonlinear best-fit regression, removing possible bias from there being ~3.5 times more flume data points. Minimum and maximum bounds were determined visually to accommodate almost all data points, assuming a consistent best-fit exponent (0.7). This approach assumes that the total observed variability in threshold at a given slope is comparable in scale to the temporal variability in threshold induced due to flow history efforts. We acknowledge that other factors may influence variability in threshold between sites, including grain size distributions (cites) and differences in relative roughness (cites). However, this approach allows the model to explore the maximum potential variability in threshold due to temporal variations in bed shear stress.*

170-200. In the results, a mix of median parameter values ($k_1$, $k_2$ etc.) for the annual best fits and mean variable values for the average best fit are used. Is there a reason for mixing the use of medians and means of parameter values? I think it might be better to use one of these consistently (median or mean parameter value) or explain why you are using medians for the annual best fits instead of means, which would be more directly comparable to the average best fit parameter values for all years combined?

Sorry for the mix-up – this was an inconsistency so thank you for catching it! We can see how that can be confusing! The differences between the means and medians are very slight, so we will go back through the document and report means for consistency in the revised version of the MS. We also recognise that some additional ambiguity may be coming from our use of "average" to describe the best-fit parameter combinations that, yield the minimum MAE when MAE is averaged across all sample years. Here we are using an average so that each year is weighted equally (as described in Line 250). We are also using a Mean Absolute Error (MAE), so admittedly we are using the terms average, mean, and median a lot in this section. To try and address/reduce some of this, we have replaced "average" with "combined" to indicate that the combined MAE is reflective of the model runs that best describe the entire dataset.

203-205. But is the lower range of values for $k_2$ and epsilon because the model is less sensitive to these parameters or could it also be because you only had three years in which weakening (which these parameters represent) was dominant, which I think you kind of imply in the discussion? Can you please clarify here?

Apologies but we're not sure what the reviewer means when they say "the lower range of values for $k_2$ and epsilon"? Best-fit values for $k_2$ and epsilon both spanned the entire parameter range. The original text reads:

*"In contrast, mean MAE values are higher and less variable when binned by $k_2$ (MAE = 0.108-0.124) and $\varepsilon$ (MAE = 0.113-0.117), suggesting that annual model performance is less sensitive to variations in these parameters (Fig. 2b,c)."*

We don't expand on this statement here as to keep our results separate from our interpretation. The previous paragraph describes the differences in median MAE for the best-performing models between strengthening and weakening years.

---

## Author Comment (AC2)

**Reviewed manuscript**: Modeling memory in gravel-bed rivers: A flow history-dependent relation for evolving thresholds of motion

**Manuscript Authors**: C. Masteller and colleagues
**Journal**: Earth Surface Dynamics
**Reviewer**: Shawn M. Chartrand

Summary and Contribution:
The authors present an empirical model which describes the discrete time evolution of the critical dimensionless Shields number (similarity-based threshold for sediment transport; note that I use "number", "condition" and "stress" below in an interchangeable way) using a unique field data set of streamflow discharge and sediment transport with the latter measured using an impact plate system installed within the Erlenbach located in the Swiss Prealps. The data are unique because sediment transport is quantified directly over annual hydrographs (along with observations of streamflow) with the impact plate system rather than using wading-type measurements combined with rating curves or transport functions for times lacking observations through measurement. As a result, the authors data permit them to identify streamflow conditions associated with the onset of bedload transport, removing substantial uncertainty in estimation of the critical dimensionless Shields number. The empirical model builds from prior work (Johnson, J.P.L., *Earth Surface Dynamics*, 2016) with revisions to the original model focused on describing the discrete time dependent behavior of the critical dimensionless Shields number in terms of so called "strengthening" and "weakening" processes. The model faithfully (and impressively) captures annual trends to the critical dimensionless Shields stress based on the available records during years when "strengthening" dominates and/or "weakening" events do not occur, with more variable performance for years when "weakening" dominates. To my understanding the authors model is the first of its kind in the geomorphic literature, and I congratulate them on their excellent work.

Based on several readings of the manuscript, I believe some effort is needed to address the comments and suggestions I raise below. I do not have any serious criticisms of the authors work, however, there are some conceptual points (bigger picture comments) that would benefit from thought and discussion in the manuscript. I think the manuscript is well written, and the figures nicely developed such that they add important illustrative dimensions and information to the narrative of the text. I appreciate the authors inclusion of discussion elements which add some critical review of the proposed empirical model making it clear that they have thought about their approach and results from a variety of viewpoints. Some specific examples of the points raised below include consideration of bedload transport in rivers as a "nonlocal" process which in part helps set the context of the measurement system as well as the fact that the authors calculation approach mixes time and space considerations, and additional discussion focused on the nuances around the interdependence of (in-channel) sediment supply and the critical dimensionless Shields condition. I also have a few questions around the parameter *Beta* that would benefit from clarification, most importantly, I cannot recover a (0,1) behavior of *Beta* with the form of Eq. 2 presented in the manuscript (I get negative values).

Thank you for the thoughtful review. We address these points in detail below.

I want to thank the journal editors and authors for the opportunity to review the submitted manuscript, and for your patience as I completed my review. I hope that my comments are helpful to the authors.

**Bigger Picture Comments (in no particular order)**

1. **Transport as a nonlocal process:** There has been important work completed that illustrates how bedload transport in rivers is a non-local process, i.e. that the transport (activity in the case of the

authors work) of particles close to the bed surface as measured at any particular point along a river profile *x* and at some time *t* is a function of transport processes upstream of *x* (within some finite distance set by the *pdf* of particle travel distances) and for some finite time interval prior and leading up to *t* (within some finite time interval set by the *pdf* of particle travel times; see Furbish et al., *GBR*, 2017 for a clear discussion; see Foufoula-Georgiou and Stark, *JGR*, 2010 for a broader discussion). The implications of bedload transport as a nonlocal process has relevance to the authors work in a number of ways: (1) it provides context for interpreting the time series of particle impacts at the Erlenbach measurement site [and hence time dependency of the dimensionless Shields stress and the critical dimensionless Shields stress] by offering a conceptually useful way to understand the authors "noisy transport data" as more of an expected outcome based on the explicit dependence of transport on time and space; (2) it offers a broader perspective to conclusions reached regarding deterministic variability of threshold evolution; (3) it elaborates the context for the authors concept of "memory" in that nonlocality provides a more concrete way to frame the authors proposal that memory "integrates the effects of the past history of both flow conditions and channel bed conditions" (lines 253-254, page 12 of the manuscript; although I also recognize that the authors idea of memory involves much longer time scales); and (4) it helps to better frame the authors calibration and application of Eq. 1 to the Erlenbach data because the authors approach mixes space and time effects, and a nonlocal perspective naturally reflects these two aspects of transport processes. In summary, the authors relate time variations of the critical dimensionless Shields number to conditions upstream of the point of observation. A nonlocal perspective of bedload transport can help the authors, to some degree, make these points in more concrete ways.

We added a paragraph near the end of the discussion section that brings up implications of nonlocality for the points the author raises, including understanding variability in threshold of motion evolution, how our model is deterministic, and how future work should explore the combination of local and nonlocal influences on "memory" encoded in thresholds of motion:

"At the same time, nonlocal controls on sediment transport will fundamentally limit how well local discharge alone can explain local sediment transport rate. Previous work demonstrates that bedload transport is a nonlocal process, because at a given location the flux reflects not only local conditions, but also spatial and temporal variations in upstream flow, sediment supply, and transport rate (e.g., Foufoula-Georgiou and Stark, 2010; Furbish et al., 2017; Martin et al., 2012). Variability from nonlocality limits the accuracy of all models for calculating bedload flux at a specific location based on local shear stress, and is not unique to our threshold evolution equation. Local flux models also cannot capture spatial and temporal grain dispersion which is as important as advection for understanding bedload transport through river networks and responses to perturbations (e.g., Bradley, 2017; Pretzlav et al, 2021; Fan et al., 2016). Our model could be applied to better determine the extent to which threshold variability (and associated transport rate variability) is a deterministic function of discharge (as Equation 1 attempts to represent), and how much of the local transport rate signal is stochastic variability, influenced by a variety of interrelated factors including nonlocality and sediment supply. Future work could also explore how the "memory" of past conditions at a given location, imperfectly encoded in $\tau_c^*$, depends on both local discharge variability and nonlocal supply effects."

2. **Equation 1 and *Beta***: In section two the authors discuss their model for the time evolution of the critical dimensionless Shields condition. I think this section will benefit from more direct discussion of the development part of the model. As it stands the section explains the components of the model

and why specific elements, etc. are justified, but their discussion does not provide much detail on the model development side. For example, if someone else attempted to recreate the authors model working from the existing text and supplemental information alone, do they have enough information to guide their thinking and decision making to eventually lead them to the form of the model presented in Eq. 1? Basically, what were the key steps or decisions made that led the authors to the present model formulation. To be clear, I am not suggesting that the authors list out every decision made along the way to the formulation of Eq. 1, but rather they provide enough information to better understand the key steps in getting there. Perhaps this can be addressed by prefacing the presentation of Eq. 1 by stating explicit hypotheses or specific ideas that underpin the authors thinking (this is briefly done at the end of section one but I am suggesting it is done in relation to presenting the proposed empirical model). Among other reasons, providing more clarity around model development will help future readers as they attempt to apply and test the model to any particular circumstance. With this in mind, it may be helpful to write out the discretized form of Eq. 1, or specify the time marching components of Eq. 1 with notation so it is clear how the initial and time dependent values are specified in the calculation procedure.

We have addressed this by both rearranging the section, and adding text to more completely explain why we picked the sigmoidal and power-law functions for the strengthening and weakening terms. We moved text from the end of the section to right after equation 1 is first presented, that the reviewer describes as giving our ideas underpinning the model. In particular, the expanded and rearranged paragraphs are these:

> "The strengthening and weakening terms combine to cause increases and decreases in $\partial \tau_c^* / \partial t$ (Eq. 1; Fig. 1). The strengthening term is generally sigmoidal for $\gamma > 1$; it goes to zero as $\tau^*$ approaches zero, and asymptotes to a value of $k_1 B$ for $\tau^*/\tau_c^* \gg 1$ (Fig. 1a). We chose a sigmoidal form in order to have strengthening over a wide range of flows, but also to limit the amount of incremental strengthening that can result from changes in grain organization at higher transport capacities. When $\tau^*/\tau_c^* < 1$, flow causes the bed to become stronger but not weaker, consistent with previous observations (Haynes & Pender, 2005; Masteller et al., 2019; Masteller & Finnegan, 2017; Monteith & Pender, 2005; Ockelford et al., 2019). Strengthening increases as $\tau^*/\tau_c^*$ approaches 1, consistent with some (Paphitis & Collins, 2005), but not all previous work (Haynes & Pender, 2007). Strengthening increases further for $\tau^*/\tau_c^* > 1$, consistent with protrusion-dependent thresholds (Masteller and Finnegan, 2017, Yager et al., 2018; Masteller et al., 2019), and with coarse grain clustering, which increases bed stability and requires transport to develop (Brayshaw, 1985; Church et al., 1998; Hassan et al., 2020; Johnson, 2017; Strom et al., 2004).
>
> At the same time, as $\tau^*/\tau_c^*$ exceeds 1, the weakening term becomes increasingly important (Fig. 1). In the absence of other constraints on the functional form of weakening with increasing $\tau^*/\tau_c^*$, we chose a power-law relation for simplicity. It seems likely to us that beds rapidly lose their strength as transport rate increases and grains are no longer interlocked through intergranular friction (Yager et al., 2018). Higher shear stresses capable of mobilizing more sediment grains can destabilize a larger fraction of the bed. Impacts from transported grains may also directly contribute to destabilization (Ancey & Heyman, 2014; Heyman et al., 2014; Lee & Jerolmack, 2018; Martin et al., 2014). Nonetheless, we note that the model is agnostic towards any specific processes driving strengthening and weakening. The combination of terms results in the transition from strengthening to weakening occurring at different $\tau^*/\tau_c^*$, depending on $\gamma$, $\varepsilon$, $k_1$, $k_2$, and $\tau_c^*$ (Fig. 1b)."

The present text lists *Beta* (Eq. 2) as a ratio of the differences between the max/min critical dimensionless Shields stress and the time dependent critical dimensionless Shields stress, respectively. Based on my reading of the manuscript I have assumed that the authors used max and min values of 0.36 and 0.036, respectively, for all associated calculations (Fig. 1 caption). However, I am not sure if this is the case. If this is true and using the form of *Beta* given in Eq. 2, it seems that *Beta* should be negative at times when the critical dimensionless stress > the min dimensionless critical Shields condition (0.036?), and < the max dimensionless critical Shields condition (0.36?)-- for example if the critical dimensionless Shields number has a value of 0.05. However, the authors state that *Beta* ranges from a value of 0-1 (lines 99-100). What am I missing? What were the values of the max and min dimensionless critical Shields stress used in the calculations of Eqs. 1 and 2? Did they change in time? Why is the authors form of *Beta* different from the form given by Johnson, 2016 (Johnson, J.P.L., Earth Surface Dynamics, 2016)? Were the max and min values calculated with the power law forms given after Eq. 2? If so, what was the value(s) of *S* used (based on the min/max values given in the Fig. 1 caption, the power law forms of the min/max suggest *S* ~ 0.11)? Also, based on the form of Eq. 2, it is difficult for me to imagine how *Beta* takes a value of 1 because the max and min values of the critical dimensionless Shields number by definition will never be equal. Clarification around *Beta* will be helpful.

We are very sorry. Our equation had a significant typo which we overlooked, and have now corrected. The form of Beta is exactly the same as that given by Johnson, 2016.

Last, the supplemental information provides important information related to calibration of Eq. 1. based in part on the work of Paphitis and Collins, 2005 (Paphitis and Collins, *Sedimentology*, 2005). It is important that this information is presented in the main text because it is key to calibration of the *Gamma* exponent of Eq. 1 term 1, and second because the referenced work was conducted experimentally using sand sized particles, which diverges from the Erlenbach field conditions.

We have moved the calibration and the rest of the supplementary material into the main paper, as requested.

3. **Critical Shields condition:** Conceptually, I stumble over how to disentangle the critical dimensionless Shields condition, the dimensionless Shields condition and the sediment supply. The critical dimensionless Shields condition and the dimensionless Shields condition are derived quantities that are calculated based on specific information (e.g. the authors estimation of the dimensionless Shields number and the critical dimensionless Shields number relies on an empirical rating curve relating streamflow discharge to the local average dimensional shear stress [Yager, E.M., PhD thesis, 2006]). Meaning, it is not possible to directly measure a critical dimensionless stress or Shields condition. The authors record of particle impacts removes a substantial degree of uncertainty related to when transport begins in the monitored section of the Erlenbach. However, the principal metrics are still subject to calculation. On the other hand, sediment supply delivered to some position *x* and for some time interval *t+Δt* is a tangible thing, something that given adequate technology, etc. can be measured and quasi-directly quantified (or at least approximately so) because it is a physical response. In rivers, for example, the diligent and careful use of sediment baskets and similar passive measurement apparatuses situated in the streambed can provide a reasonable record of supply magnitude, grain size composition and the rough time interval over which a basket fills (e.g. Hassan and Church, *Water Resources Research*, 2001).

This is an interesting interpretation with which we respectfully disagree, at least in part. Shear stress is a physically meaningful variable, as is sediment supply. The sediment supply to a reach is the sediment transport rate from just upstream into a given reach. In principle, it is possible to measure sediment transport rate, as the reviewer points out, it is just difficult. But it is also possible—and arguably easier— to measure shear stress. We think of dimensionless shear stress, and critical dimensionless shear stress, as physically meaningful measurements as well. Shear stress is the product of slope and flow depth (or hydraulic radius). Slope is constant, so flow depth is the main driver for variability. Depth can be both precisely and accurately measured, as can the onset of particle impacts. Respectfully to the reviewer, the installation of impact plates at the Erlenbach, and associated measurement capabilities for flow depth and mass flux measurements, makes it a state-of-the-art measurement facility. Ample, diligent work has been done by multiple authors of this article to reliably calibrate impact plate records to bedload flux and grain size composition. It is correct that at the Erlenbach the impact plates are positioned in a cross-section slightly downstream of the natural channel but work from the authors has demonstrated that a typical travel time for a bedload particle from the natural bed to the geophone plates is about 30s (which is less than the measurement period). Based on this previous work, we emphasize that the Erlenbach impact plate and discharge records, and by extension, our calculations of critical Shields stress, represents both a measurable difference in sediment transportability at the Erlenbach, but also a physically meaningful different in the threshold for motion.

At several locations in the manuscript the authors state or use other studies to suggest that sediment supply influences threshold evolution (lines 43-44), and in turn that supply depends on thresholds (lines 58-60).

We reworded the sentence at lines 58-60; we were not trying to say that sediment supply depends on thresholds (rather, thresholds depend on the supply). (That said, we do not think it is necessarily wrong, as the sediment supply into a given reach depends on the integrated sediment transport rate from upstream, which in turn depends on the distribution of transport thresholds upstream. But, this is not the focus of this part of the introduction).

I understand conceptually how thresholds and sediment supply are inter-related, noting that "sediment supply" can mean a couple different things—i.e. in-channel storage, hillslope derived, and so forth. It would be helpful if the authors
developed an expanded discussion of sediment supply vs. thresholds in which they more carefully step through the nuances of how these are inter-related and inter-dependent, and what is explicitly meant by sediment supply in the context of the manuscript. The last paragraph of section six could be suitable to expand the discussion, although it would be more impactful to have this presented in section one as the reader will have a clearer picture when reading the remainder of the manuscript.

We modified both the introduction and the discussion section to address this point, attempting to better clarify how discharge and sediment supply both influence threshold evolution. First, to make it more directly clear how the threshold parameter gets used to calculate bedload transport rate, we added the classic Meyer-Peter and Muller equation and use it to illustrate how the threshold parameter is a lumped measure to account for all factors other than fluid forcing that influence transport rate, including both discharge variability and sediment supply. Second, we rearranged and added introduction text to more obviously separate citations of previous work describing threshold changes with discharge and threshold changes with sediment supply. Third, near the end of the section we now describe how timeseries of discharge and shear stress are easier to constrain than upstream sediment supply to a given reach, and also how the sediment supplied to a given river reach is not independent of discharge,

since that supply is sediment transported by flow in the river. We added the citation to Hassan and Church, 2001.  Finally, we reworded to make it even more clear that our goal in the present analysis is to explore how well discharge variability alone can explain threshold evolution.

In the discussion, we expanded our thoughts on sediment supply controls with this edited paragraph:

"Perhaps the biggest mechanistic limitation of our model is that it only accounts for discharge controls on evolving thresholds, even though sediment supply has also been shown to explicitly influence transport rates in the Erlenbach data (Turowski et al., 2011, Rickenmann, 2020; 2024). In flumes, it is straightforward to impose the upstream sediment supply, measure the flux exiting the flume, and simultaneously measure changes along the flume bed, allowing thresholds to be evaluated through time as a function of supply (e.g., Johnson, 2016). We are unaware of field monitoring sites that directly measure comparable timeseries of transport data in sequential channel reaches, making it difficult to directly isolate supply controls on threshold evolution in field settings. Some sources of sediment supply into a given channel reach, such as shallow landslides a short distance upstream not triggered by precipitation, may be uncorrelated with channel discharge. However, the timing and magnitude of many processes that supply sediment to channels, such as bank failures, debris flows, and shallow landslides driven by very recent precipitation, are likely correlated with timeseries of channel discharge. In addition, sediment supplied from farther upstream in a watershed is transported into a given reach by channel flow. Seasonal trends in supply, such as from increased activity of hillslope processes during winter months followed by subsequent snowmelt or storm flow that evacuates the sediment (e.g., Moog and Whiting, 1998; Mao et al, 2014), are not controlled by discharge but nonetheless may correlate with cumulative discharge as gravel is transported through the channel (e.g., Pretzlav et al., 2020). Therefore, discharge timeseries may be able to implicitly account some temporal variations in local supply, and therefore possibly be able to explain some supply-dependent $\tau_c^*$ variability. The degree of correlation between supply timeseries and discharge timeseries would likely vary among watersheds based on dominant processes. Future work should attempt to disentangle how sediment supply influences our parameter calibrations."

4. **Bedload transport and threshold conditions:** There are discussion elements which frame bedload transport around threshold conditions, inter-connections between these conditions and gravel-bed river geometry (e.g. lines 22-23; lines 222-224) and the nature of threshold conditions during floods or transporting events (e.g. lines 22-23, lines 38-39, lines58-61). In several cases there is important literature missing from the discussion. For example, ideas around bedload transport, transporting floods and gravel-bed river geometry have been the subject of significant field-based data collection efforts, some of which provide results not as definitive as that suggested in the present form of the manuscript. Whiting et al., 1999 (Whiting et al., *GSAB*, 1999) present a comprehensive dataset based on hundreds of bedload measurements across more than 10 headwater rivers which suggests that gravel-bed river geometry is maintained in the Idaho batholith (a snowmelt dominated system) at flows less then bankfull (~0.80 bankfull), with the common 1- to 2-orders of magnitude variability in the sediment-flow rating curves. There are many other examples in the literature as well (that I know the authors are aware) which suggest that bankfull flow is not necessarily the most important flow in maintaining channel form (geometry). I raise this point because the subject is the matter of significant debate in fluvial geomorphology, and presenting a more balanced picture of what the literature

suggests seems appropriate.

We respectfully note that the focus of the manuscript is on evolving thresholds of motion, not channel geometry, morphodynamic feedbacks, or any analyses of bankfull vs effective discharges. We briefly mention in a small number of places that gravel-bed rivers can be modeled as threshold channels where the threshold of motion is an important control on channel morphology, but do not get into any real discussions of the significance of bankfull flows, or flows that are close but not quite bankfull.

In the discussion, we added a citation to Whiting et al. (1999) and also to Emmett and Wolman (2001), and de-emphasized the importance of bankfull flow to river geometry. The new wording is this:

"Sediment is transported infrequently in gravel-bed rivers because of $\tau_c^*$. Much transport occurring during at conditions just exceeding the threshold of motion during discharges that are often relatively close to bankfull (e.g., Emmett and Wolman, 2001; Parker 1978, Phillips and Jerolmack, 2016; Pretzlav et al., 2020; Whiting et al., 1999)."

Minor and Editorial Comments

**Lines 16-17 (comment also relates to lines 237-238).** The sequence, timing and magnitude of significant precipitation events and heatwaves (both of which cause floods) are stochastic. Because "weakening" events are directly linked with floods, the authors conclusion that weakening events are more stochastic than strengthening ones is clear. Use of the phrase "...suggests that flood-induced bed weakening is more stochastic and less predictable then strengthening." is a little confusing. Are there additional mechanisms not related to flood events (this would include mass movements, etc.) that could cause weakening? I guess I am unsure whether this is a surprise or unexpected from the authors point of view? I think the authors specifying "more stochastic and less predictable" is what is causing my confusion. Also, what does "more stochastic" mean?

We are trying to express the variability in terms of sediment transport, not the hydrologic processes. How much transport thresholds change, and how predictable the magnitude of those changes are, may be more variable and therefore harder to predict within some level of tolerance or accuracy, than strengthening changes. This is separate from floods themselves being harder to predict than base flow.

We changed the wording to hopefully better express this, while still keeping the wording short and focused in the abstract: "...magnitudes of bed weakening may be more variable and difficult to accurately predict as a function of flood characteristics than strengthening during lower flows."

**Lines 22-23.** What do "close" and "floods" mean? Can the authors be more specific?

We changed the wording to be more specific: "Bedload transport often occurs at shear stresses only slightly higher than threshold conditions even during large floods…"

**Lines 23-26.** Blom et al., 2017 conclude that the influence of climate change (and hence extreme floods) to river geometry in the zone downstream of the hydrograph boundary layer and upstream of the terminal backwater zone may be negligible (page 19 of Blom et al., 2017). How does this fit into the concept of mapping "climate onto fluvial processes"?

In this one motivating sentence we tried to summarize how the literature we cite here, including (but not limited to) Blom et al., describe a range of ways in which hydrograph variability, which is influenced by climate, in turn influences bedload transport and channel evolution. We think that the arguments of Blom et al. (2017) that the reviewer mentions are reasonably described as "the relative importance of extreme events for channel evolution" (our original wording).  Nonetheless, to clarify the reviewer's point, we changed the wording to "the relative importance (or unimportance) of extreme events for channel evolution".

**Lines 27-29**. I had to read this sentence a few times to understand it. Can the authors re-phrase for clarity?

We rewrote the sentence to have a simpler structure and to be clearer.

**Lines 57. Minor point:** Do all sediment pulses cause disequilibrium? Sediment pulses are commonly of short relative timescales. Ideas around disequilibrium can be associated with longer relative timescales. Pulses can disrupt local bed elevations, grain size populations and hence local transport rates—this fits at least two ideas of disequilibrium. But pulses can also fall under the concept of dynamic equilibrium. I am wondering if/how disequilibrium as used in this sentence differs or is similar to the use of the same word in the following sentence?

So as to avoid going "down a rabbit hole" of explaining different meanings and usages of transport disequilibrium, we reworded the parts of the introduction that used the term. We feel like explaining it would be a digression that would take away from our focus on thresholds.

**Lines 58-70.** I think the work of Moog and Whiting, 1998 (the authors include this work in their references) is relevant to the discussion here. The key from Moog and Whiting relates to hysteresis and their data which suggest that prior to the occurrence of the estimated transporting flow each season, there was higher bedload transport for a given flow than afterward. This trend was attributed to limitation or exhaustion of in-channel sediment supply as the snowmelt hydrograph progressed. This comment in part relates to my "bigger picture" comment above related to sediment supply and the Shields condition.

We now cite Moog and Whiting (1998) in the previous paragraph while talking about sediment supply and transport thresholds.

**Lines 165-166.** How do you know that certain data are "outliers"? Does the concept of an outlier make sense in an inherently "noisy" system? Even time series of flux or transport in the most simple of experiments is, for example, noisy (for example, see Fig. 6 of Ancey et al., *Physical Review E*, 2006).

We changed the wording from "outliers in rather noisy transport data" to "field-based data points which exhibit large amounts of scatter". We were not trying to imply that we omitted any individual points (outliers) from the analysis (we did not).

**Lines 179-186 – Figure 2.** Nice figure with lots of great information. What do the black dots in the lower panels represent? I read carefully and could not find mention of what they represent.

Thanks for this.  We added the following line to the figure caption (now figure 3):  "Comparison of field data (black dots, representing $\tau_c^*$ at the start of each transporting event) to the best-fit model(s). "

**Lines 219.** "…higher *relative* transport capacities"?

We changed the sentence to better reflect what we were trying to say: "The transport capacity ($\tau^*/\tau_c^*$) at which the model terms combine to transition from overall strengthening to weakening varies for different parameter combinations (Figure 1)."

**Lines 228.** There is a missing word towards the end of the line.

We reworded the sentence.

**Lines 257-258.** The sequence, timing and magnitude of future floods is a stochastic phenomenon dependent on future climate conditions, etc. I believe it is generally held that future conditions can be "projected" (not predicted) when there is a dependence on climate-related phenomena because of the probabilistic nature of the problem. Perhaps I misunderstand the intent of the sentence?

Looking at the sentence in question in isolation ("by knowing $\tau_c^*$, one can predict future channel response to floods"), we can see how it is a stronger statement than we intended. We have reworded to "For the Erlenbach, our results using the calibrated model demonstrate that knowing τ_c^* prior to a given flood improves the prediction of transport during that flood."

**Lines 258-262.** I encourage the authors to provide a more comprehensive discussion of how their model of the critical dimensionless Shields stress can be applied in other circumstances, with particular details related to what data, at a minimum, are necessary to locally calibrate their Eq. 1. I think data additional to a high-resolution discharge time series is necessary.

We have added a paragraph that expands on how we envision applying the model to other data sets. We were not trying to imply that a discharge timeseries was the only data required for a calibration. The new text says (in part):

> "Reach-averaged starting values of τ_c^* could be estimated based on bed grain size and bankfull geometry. Additional model calibration will vary depending on the intended application of the equation. To characterize long-term variability in τ_c^*, calibration of the model over approximately 30 transport events may be needed to reliably capture the expected variability of τ_c^*, assuming a normal distribution, as observed at the Erlenbach by Masteller et al. (2019). However, this commonly used minimum sample size assumes independent observations, which does not apply here. An alternative approach is to calibrate the model based on the number of subsequent events over which τ_c^* remains correlated. Masteller et al., (2019) also found a loss of correlation between τ_c^* values after 10-13 transport events. This number of events may be sufficient to calibrate the model to capture the trajectory of τ_c^* over time. We recognize that requiring 10–30 measurements of τ_c^* observations may not always be feasible. Future studies should assess the necessary level of calibration for different applications."

**Lines 263-265.** I don't understand how "transport disequilibrium" influences transport rates? My understanding of the idea is that transport disequilibrium depends on how observed transport compares to calculated transport (Rickenmann, D., *Water Resources Research*, 2020).

For simplicity and to avoid a digression into definitions of transport disequilibrium, we removed those words from the sentence; now it says this: "Perhaps the biggest mechanistic limitation of our model is that it only accounts for discharge controls on evolving thresholds, even though sediment supply has also been shown to explicitly influence transport rates in the Erlenbach data (Rickenmann, 2020; 2024)."

To answer the question, we were intending transport disequilibrium in the same way that Johnson (2016) used the term: Equilibrium transport in a given reach means that the flux in and out of the reach are matched; this is the equilibrium condition that reaches tend towards (e.g., Mackin, 1948, Concept of the Graded River). Disequilibrium transport implies net erosion or deposition in a given reach. Rickenmann defined the "disequilibrium ratio" in the way the reviewer describes.

**References not in the manuscript**:
Ancey, C., Böhm, T., Jodeau, M., and Frey, P.: Statistical description of sediment transport experiments, Physical Review E, 74, 11302–11302, https://doi.org/10.1103/PhysRevE.74.011302, 2006.

Foufoula-Georgiou, E., and Stark, C.: Introduction to special section on Stochastic Transport and Emergent Scaling on Earth's Surface: Rethinking geomorphic transport—Stochastic theories, broad scales of motion and nonlocality, *J. Geophys. Res.*, 115, F00A01, doi:10.1029/2010JF001661, 2010.

Furbish, D. J., Fathel, S. L., and Schmeeckle, M. L.: Particle Motions and Bedload Theory: The Entrainment Forms of the Flux and the Exner Equation, in: Gravel-bed rivers, Wiley-Blackwell, 97–120, https://doi.org/10.1002/9781118971437.ch4, 2017.

Hassan, M. A., and M. Church: Sensitivity of bed load transport in Harris Creek: Seasonal and spatial variation over a cobble-gravel bar, *Water Resour. Res.*, 37(3), 813–825, doi:10.1029/2000WR900346, 2001.